# EVALUATING MODEL-BASED PLANNING AND PLAN-NER AMORTIZATION FOR CONTINUOUS CONTROL

**Arunkumar Byravan**[*†]  **Leonard Hasenclever**[*†]  **Piotr Trochim**[*]  **Mehdi Mirza**[*]

**Alessandro Davide Ialongo**[‡§]  **Yuval Tassa**[*]  **Jost Tobias Springenberg**[*]

**Abbas Abdolmaleki**[*]  **Nicolas Heess**[*]  **Josh Merel**[‡¶]  **Martin Riedmiller**[*]

## ABSTRACT

There is a widespread intuition that model-based control methods should be able to surpass the data efficiency of model-free approaches. In this paper we attempt to evaluate this intuition on various challenging locomotion tasks. We take a hybrid approach, combining model predictive control (MPC) with a learned model and model-free policy learning; the learned policy serves as a proposal for MPC. We find that well-tuned model-free agents are strong baselines even for high DoF control problems but MPC with learned proposal distributions and models (both trained on the fly or transferred from related tasks) can significantly improve performance and data efficiency in hard multi-task/multi-goal settings. Finally, we show that it is possible to distil a model-based planner into a policy that amortizes the planning computation without any loss of performance. Videos of agents performing different tasks can be seen on our website.

## 1 INTRODUCTION

In recent years, model-free RL algorithms have improved and scaled to the point where it is feasible to learn adaptive behavior for high-dimensional systems in diverse settings (Schulman et al., 2017; Heess et al., 2017). Despite these improvements, there is a widespread intuition that model-based methods can further improve data efficiency. This has led to recent advances that demonstrate improved learning efficiency by leveraging model learning during policy training (Lowrey et al., 2019; Nagabandi et al., 2020; Hubert et al., 2021). However, there is also work which urges moderation in interpreting these results, by showing that well-tuned model-free baselines can compare favorably against some model-based approaches (Springenberg et al., 2020; van Hasselt et al., 2019).

In this paper, we focus on model-based RL with learned models and proposals.[1] We use model-predictive control (MPC) to improve the quality of behavior generated from a policy that serves as a proposal for the planning procedure. When acting, the MPC-based search procedure improves the quality of the data collected which serves as a sort of "active exploration". This data can then be used to improve the proposal, in effect consolidating the gains. This hybrid approach interpolates between model-free RL, on one side, and trajectory optimization with a model on the other.

Fundamentally, the spectrum on which this hybrid approach is situated reflects a trade-off in terms of reusability / generality versus compute cost at deployment time. Policies obtained by model-free RL often generalize poorly outside of the situations they have been trained for, but are efficient to execute (i.e., they are fully amortized). Models offer potentially greater generalization, insofar as the model is accurate over a broad domain of states, but it can be computationally costly to derive

---

[*]DeepMind

[†]Equal contributions. Correspondence to {abyravan, leonardh}@google.com

[‡]Work done at DeepMind

[§]Max Planck Institute for Intelligent Systems, Tübingen, Germany and Computational and Biological Learning Group, University of Cambridge

[¶]Facebook Reality Labs

[1]We use the term proposal to mean a "proposal distribution", which is a distribution over the space of actions.

actions from models. Ultimately, the pure planning approach involving deriving actions from models is usually prohibitive and some form of additional knowledge is required to render the search space tractable for real-time settings. Coming from the model-free RL perspective, the natural approach is to learn a policy that serves as a relatively assumption-free proposal for the planning process.

While in most model-based RL publications the ambition is to speed up policy learning for a single task, we believe this is not necessarily the most promising setting, as it is difficult to dissociate the learning of a dynamics model (which is potentially task independent) from that of a value function (which is task specific). A different intuition is that model-based RL might not accelerate learning on a particular problem, but rather enable efficient behavior learning on new tasks with the same embodiment. In such transfer settings, we might hope that a dynamics model will offer complementary generalization to the policy, and we can transfer the policy, model, or both.

In this work, we validate these intuitions about the relative value of the different components of this system. For high-dimensional locomotion problems, we find that even with a good learned model successful MPC requires good proposal distributions to succeed. This effect is particularly pronounced in domains with multiple goals (or equivalently multiple different tasks). Building on this insight we show that it is sometimes possible to leverage learned models with a limited search budget to boost exploration and learn policies more efficiently. Policies that are trained by off-policy updates from data acquired through planning do not reliably perform well when used without planning in our experiments. To overcome this problem, and enable planner amortization into a policy that does not require planning at test time, we propose training the policy with a combination of a *behavioral cloning* objective (on MPC data) and an off-policy update with MPO (Abdolmaleki et al., 2018b). When transferring models and proposals to other tasks we find only marginal improvements in data efficiency relative to a well-tuned model-free baseline. Overall, our results suggest that while MPC with learned models can lead to more data efficiency, and planners can be amortized effectively into compact policies, it is not a silver bullet and model-free methods are strong baselines.

## 2 Background & related work

**Model-predictive control** refers to the use of model-based search or planning over a short horizon for selecting an action. In order to make it computationally tractable, it is common to "seed" the search at a given timestep with either an action from a proposal policy or an action obtained by warm-starting from the result of the previous timestep. In simulation, it has been demonstrated that a fairly generic MPC implementation can be effective for control of a relatively high DoF humanoid (Tassa et al., 2012) and that MPC with learned models (Chua et al., 2018; Wang & Ba, 2020; Nagabandi et al., 2020) can achieve data-efficient solutions to simple control problems. Real RC helicopter control has also been achieved using an MPC approach that made use of a learned model (Abbeel et al., 2006). MPC approaches are gaining wider use in robotics (multicopters (Neunert et al., 2016; Torrente et al., 2021), quadruped (Grandia et al., 2019; Sleiman et al., 2021), humanoids (Tassa et al., 2014; Kuindersma et al., 2016)), and dexterous manipulation (Nagabandi et al., 2020); but the computational speed of the planner is a bottleneck for hardware deployment. There are different ways around this, with the core practical solution being to plan in lower dimensional reduced coordinate models. Alternatively, POPLIN (Wang & Ba, 2020) explores learning proposals for MPC and planner distillation but is tested on simple tasks and does not leverage model-free RL for proposal learning, and LOOP (Sikchi et al., 2022) uses an off-policy actor to control exploration for MPC.

**Amortized policies** map states to actions quickly, but implicitly reflect considerable knowledge by being trained on diverse data. Model-free RL produces policies which amortize the knowledge reflected in rollouts required to produce them. Similarly, it is possible to produce diverse trajectories from a planner and distil them into a single policy (Levine & Koltun, 2013; Mordatch & Todorov, 2014; Mordatch et al., 2015; Pan et al., 2017; Xie et al., 2021).

**Reinforcement learning approaches with MPC** have become more popular recently, and our work fits within this space. As noted in the introduction, previous work often emphasizes the role of amortization through the learning of value functions and models. TreePI (Springenberg et al., 2020), MuZero (Schrittwieser et al., 2020; Hubert et al., 2021), SAVE (Hamrick et al., 2020), DPI (Sun et al., 2018) and MPC-Net (Carius et al., 2020) all perform versions of hybrid learning with model-based MCTS or planning being used to gather data which is then used to train the model and value function to accelerate learning. Other recently proposed algorithmic innovations blend MPC with learned

value estimates to trade off model and value errors (Bhardwaj et al., 2021). Here, we primarily consider learning dynamics models to enable transfer to new settings with different reward functions.

**Other uses of models** unlike hybrid MPC-RL schemes have also been explored in the literature; however, they are not the focus of this work. Nevertheless we highlight two of these approaches: value gradients can be backpropagated through dynamics models to improve credit assignment (Nguyen & Widrow, 1990; Heess et al., 2015; Amos et al., 2020) and it is possible to train policies on model rollouts to improve data efficiency (Sutton, 1991; Janner et al., 2019). Recently, there has also been considerable effort towards exploring model-based offline RL methods in order to gain full value from offline datasets (Yu et al., 2020; Argenson & Dulac-Arnold, 2021; Kidambi et al., 2020).

## 3 TASKS

In this paper we consider a number of locomotion tasks of varying complexity, simulated with the MuJoCo (Todorov et al., 2012) physics simulator. We consider two embodiments: an 8-DoF ant from `dm_control` (Tunyasuvunakool et al., 2020) and a model of the Robotis OP3 robot with 20 degrees of freedom. For each embodiment, we consider three tasks: walking forward, walking backward and "go-to-target-pose" (GTTP), a challenging task that is the focus of our evaluation. In all tasks, the agent receives egocentric proprioceptive observations and additional task observations.

In the walking forward and backward tasks the agent is rewarded for maximizing forward (or backward) velocity in the direction of a narrow corridor. For the OP3 robot we also include a small pose regularization term. The task is specified through a relative target direction observation.

The GTTP task, which builds on existing motion tracking infrastructure(Hasenclever et al., 2020), consists of either body on a plane, with a target pose in relative coordinates as a task-specific observation and proximity to the target pose rewarded. When the agent is within a threshold distance of the target pose (i.e. it achieves the current target), it gets a sparse reward and a new target is sampled. For the ant we use target poses

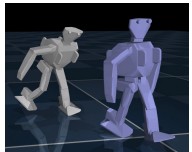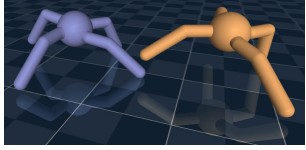

Figure 1: Go-to-target-pose (GTTP) task with the OP3 & Ant. The agent has to reach the target pose (blue).

from policies trained on a standard go-to-target task (Tunyasuvunakool et al., 2020). For the OP3, we use poses from the CMU mocap database (cmu) (retargeted to the robot). We use thousands of different target poses; the agent has to learn to transition between them. Thus the GTTP task can be seen as either a single highly diverse task or as a multi-task setting with strongly related tasks and consistent dynamics. We believe the GTTP task should be particularly amenable to model-based methods: it combines a high-dimensional control problem with a diverse goal distribution. This makes it hard to solve quickly with model-free methods. However, since all goals share the same dynamics, a dynamics model should help leverage the common structure. See Sec. A for an in-depth motivation and Sec. F.5 for results on tasks from the DeepMind Control Suite (Tassa et al., 2018).

## 4 METHOD

### 4.1 MODEL PREDICTIVE CONTROL (MPC) FOR BEHAVIOR GENERATION

We consider control obtained by model-based planning that refines initial action-samples from a proposal distribution. MPC executes the first action from the planned action sequence and iteratively re-plans after each timestep (c.f. actor loop in Alg. 1). In this scheme, actions are sampled either from the planner (PLANNER) with probability $p_{plan}$ or the proposal $\pi_\theta$ (with parameters $\theta$) with probability $1 - p_{plan}$. Setting $p_{plan} = 0$ results in using only samples from the proposal and vice-versa for $p_{plan} = 1$. Setting $p_{plan} = 0.5$ leads to interleaved execution of proposal and planner actions, like the stochastic mixing of student and expert actions in DAGGER(Ross et al., 2011).

The PLANNER subroutine takes in the current state $s_t$, the proposal $\pi_\theta$ (with parameters $\theta$), a model $m_\phi$ (with parameters $\phi$) that predicts next state $s_{t+1}$ given current state $s_t$ and action $a_t$, the known reward function $r(s_t, a_t, s_{t+1})$ and, optionally, a learned state-value function $V_\psi(s)$ (with parameters $\psi$) that predicts the return given state $s$. While we consider primarily a Sequential Monte Carlo based planner (SMC, Alg. 2) (Piché et al., 2019; Gordon et al., 1993) and the Cross-Entropy Method (CEM, Alg. 3) (Botev et al., 2013), other planners (Williams et al., 2017) could be used. See Sec. B.

---

**Algorithm 1** Agent combining MPC with model-free RL

---

**Given:** Randomly initialized proposal $\pi_\theta$, (pre-trained or random) model $m_\phi$, random critic $Q_\psi$, optionally (pre-trained or random) task-agnostic proposal $\rho_\omega$. *// Modules to be learned*
**Given:** Known reward $r$, planning probability $p_{plan}$, replay buffer $\mathcal{B}$, MPO loss weight $\alpha$, BC loss weight $\beta$, learning rates & optimizers (ADAM) for the different modules. *// Known modules and parameters*

*// MPC loop – Asynchronously on the actors*
**while** True **do**
    Initialize ENV and observe state $s_0$.
    **while** episode is not terminated **do**
        *// Choose between planner and proposal action depending on $p_{plan}$*
        *// Use mixture of (pre-trained) task-agnostic proposal ($\rho_\omega$) and learned proposal ($\pi_\theta$) as proposal for proposal transfer exps (Sec. 5.4). Use pre-trained model for model transfer exps (Sec. 5.3).*
        $a_t \sim \begin{cases} \text{PLANNER}(s_t, \pi_\theta, m_\phi, r, V_\psi) & \text{if } x \leq p_{plan} \\ \pi_\theta(s_t) & \text{otherwise.} \end{cases}$    where $x \sim U[0,1]$
        Step ENV$(s_t, a_t) \rightarrow (s_{t+1}, r_t)$ and write transition to replay buffer $\mathcal{B}$
    **end while**
**end while**

*// Asynchronously on the learner*
**while** True **do**
    Sample batch $B$ of trajectories, each of sequence length $T$ from the replay buffer $\mathcal{B}$
    Update action-value function $Q_\psi$ based on $B$ using Retrace (Munos et al., 2016).
    Update model $m_\phi$ based on $B$ using multi-step loss in Equation 20 (Sec. 4.2)
    Update proposal $\pi_\theta$ based on $B$ using Equation 4 (Sec. 4.3)
    Optionally, for from-scratch experiments, update task-agnostic proposal $\rho_\omega$ using behavioural cloning (Equation 1) on transitions in $B$.
**end while**

---

The hybrid agent in Alg. 1 has several learnable components: the model $m_\phi$, the proposal $\pi_\theta$ as well as the (optional) value function $V_\psi$; these are learned based on data generated by the MPC loop on the actor and we measure learning progress w.r.t environment steps collected. Note that we assume the reward function is known since it is under the control of the practitioner in most robotics applications.

## 4.2 LEARNING DYNAMICS MODELS

Successfully applying MPC requires learning good dynamics models. In this paper, we train predictive models $m_\phi$ that take in the current state $s_t$ and action $a_t$ and predict the next state $s_{t+1} = m_\phi(s_t, a_t)$. This model has both learned and hand-designed components; the task-agnostic proprioceptive observations are predicted via "black-box" neural networks while any non-proprioceptive – task-specific – observations (e.g. relative pose of the target) are calculated in closed form from the predicted proprioceptive observations together with a known kinematic model of the robot. The learned components are parameterized as a set of deterministic feed-forward MLPs, one per observation group (e.g. joint positions), and predict the next observation from the current observations and action. This model is trained end-to-end via a multi-step squared error (Eqn. 20) between an open loop sequence of predicted states and true states sampled from the replay buffer. We use deterministic models as they are more amenable to multi-step losses and an ablation study (Sec. F.4 of the supplement) as well as a recent benchmarking effort (Lutter et al., 2021) found minimal differences between using deterministic vs stochastic model ensembles for model-based planning. Training happens from scratch, concurrently with policy learning. More details are in Sec. C of the supplement.

## 4.3 LEARNING PROPOSAL DISTRIBUTIONS

As we will show, planning with a good dynamics model by itself is insufficient for solving challenging continuous control problems. That is, generating behavior according to the MPC loop in Alg. 1 with a goal-unaware proposal fails to solve the challenging GTTP tasks *even* when planning with the ground-truth model and a known reward function. This is due to the fact that the action space is large, and the planning algorithm is limited by computational constraints and a finite horizon. In order for planning to succeed, we find it vital to provide planners with an action proposal distribution, which helps search by increasing the probability of sampling plausible, task-relevant actions. In addition, we show that the learned proposal can itself be deployed without planning at test time; this can be particularly useful in compute constrained settings where planning on the robot may not be feasible.

But what constitutes a good proposal? A natural answer to this question is that a good proposal produces actions, for each state, that lead to task success, and the planner selects the best among already good candidate actions. A simple strategy to obtain such a proposal is to iteratively amortize the planning process. That is, starting from any proposal, execute the planner for a given duration to return an improved action and fit the returned action; this will sharpen the proposal and lead to the planner focusing on better actions in the next iteration. This idea has been used in the RL community before and underlies modern search based algorithms such as AlphaZero (Schrittwieser et al., 2020).

In the MPC setting we can formalize the idea as follows: let $\pi_{\mathcal{B}}(a|s)$ refer to the stochastic mixture of $\text{PLANNER}(s_t, \pi_\theta, m_\phi, r, V_\psi)$ and $\pi_\theta(a|s)$ with probability $p_{\text{plan}}$ (i.e., as actions are selected according to the MPC loop in Alg. 1). And let $\mathcal{D}_{\pi_{\mathcal{B}}}$ be a dataset of states and actions collected by executing this policy – note that in practice for data efficiency reasons we consider a dataset $\mathcal{D}_{\tilde{\pi}_{\mathcal{B}}}$, where $\tilde{\pi}_{\mathcal{B}}$ is the average behavior distribution during training, which changes over time due to the proposal and model being learned. Clearly, assuming the planning procedure produces improved actions, we expect that $\tilde{\pi}_{\mathcal{B}}$ is at least as good as the average proposal policy $\tilde{\pi}_\theta$. We can then improve our proposal by minimizing the forward KL divergence $\mathbb{E}_{s \in \mathcal{D}_{\tilde{\pi}_{\mathcal{B}}}}[KL(\tilde{\pi}_{\mathcal{B}}(\cdot|s)|\pi_\theta(\cdot|s))]$ which leads to an improving average proposal over time, and is equivalent to maximizing:

$$\mathcal{J}_{\text{BC}}(\theta) = \mathbb{E}_{s,a \in \mathcal{D}_{\tilde{\pi}_{\mathcal{B}}}}[\log \pi_\theta(a|s)], \tag{1}$$

i.e. the proposal is learned by amortizing the planner via behavioral cloning of the planner actions.

Unfortunately, in cases where the planner cannot find an improvement on the proposal (as is the case for random proposals in our experiments), the above objective may lead to premature convergence at sub-optimal behavior (see e.g. Wang & Ba (2020)). To ameliorate this issue, we can consider hybrid updates that both amortize the planner, but also directly favor actions that lead to an improvement in terms of cumulative return via an off-policy RL policy update. We will use the MPO (Abdolmaleki et al., 2018b) algorithm, although other recent RL algorithms could be used instead. More precisely, the MPO policy improvement step involves using a learned action-value function $Q^{\pi_\theta}(s,a) \approx \mathbb{E}_{\pi_\theta}[\sum_t \gamma^t r_t(s_t, a_t)|s_0 = s, a_0 = a]$, and optimizing the KL constrained RL objective:

$$\mathcal{J}_{\text{MPO}}^E(q) = \mathbb{E}_{s \in \mathcal{D}_{\tilde{\pi}_{\mathcal{B}}}}[\mathbb{E}_q[Q^{\pi_\theta}(s,a)]] \qquad \text{s.t.} \quad \mathbb{E}_{s \in \mathcal{D}_{\tilde{\pi}_{\mathcal{B}}}}[KL(q(\cdot|s)|\pi_\theta(\cdot|s))] < \epsilon. \tag{2}$$

It is known (see Abdolmaleki et al., 2018b) that the solution of this optimization is given in closed form as $q(a|s) \propto \pi_\theta(a|s) \exp(Q^{\pi_\theta}(s,a)/\eta)$, where $\eta$ is a dual variable optimized such that the KL-constraint on the policy is fulfilled. We can fit this improved policy by minimizing the KL divergence $\mathbb{E}_{s \in \mathcal{D}_{\tilde{\pi}_{\mathcal{B}}}}[KL(q(\cdot|s)|\pi_\theta(\cdot|s))]$ which is equivalent to maximizing the weighted log-likelihood:

$$\mathcal{J}_{\text{MPO}}(\theta) = \mathbb{E}_{s \in \mathcal{D}_{\tilde{\pi}_{\mathcal{B}}}}[\mathbb{E}_{\pi_{\theta'}}[\exp(Q^{\pi_{\theta'}}(s,a)/\eta) \log \pi_\theta(a|s)]], \tag{3}$$

where $\pi_{\theta'}$ is the last reference policy (fixed for this optimization). In practice MPO uses an additional trust-region constraint to stabilize learning. We can combine the two objectives (Eqns. 1 and 3) for improving the proposal via a simple weighting to obtain the complete objective

$$\mathcal{J}_{\text{MPO+BC}}(\theta) = \alpha \mathcal{J}_{\text{MPO}}(\theta) + \beta \mathcal{J}_{\text{BC}}(\theta). \tag{4}$$

The full agent showing the actor and learner loops is shown in Alg. 1. In our experiments we compare several variants corresponding to different choices of $p_{plan}$, $\alpha$ and $\beta$. See Sec. D for further details and rationale on our choice of MPO as the learning algorithm.

## 5 RESULTS

### 5.1 EVALUATING PRE-TRAINED MODELS AND PROPOSALS

In a first set of experiments we study how MPC (Alg. 1, $p_{\text{plan}} = 1$) performs on the locomotion tasks considered in the paper. We evaluate performance with two different models, the ground truth MuJoCo simulator and a pre-trained model that is trained on data from a successful agent on the corresponding task (see Sec. 4.2). We also consider two different proposal distributions: a zero-mean Gaussian proposal and a task-agnostic proposal that is pre-trained with a behavioral cloning objective on logged data from a successful agent (similar to the prior distribution in Galashov et al. (2019), see Sec. E.2). This pre-trained proposal depends only on proprioceptive information but not task specific observations, and therefore is a reasonable prior capturing average behavior. We assume a known reward function but do not use a learned value function. The best results of a large hyper

| Model | Planner | Proposal | Ant forward | backward | GTTP | OP3 forward | backward | GTTP |
|---|---|---|---|---|---|---|---|---|
| Near-Optimal performance | | | $\approx 1580$ | $\approx 1580$ | $\approx 650$ | $\approx 480$ | $\approx 570$ | $\approx 730$ |
| – | – | Gaussian | $0 \pm 1$ | $0 \pm 1$ | $197 \pm 2$ | $5 \pm 1$ | $-14 \pm 1$ | $9 \pm 1$ |
| ground truth | CEM | Gaussian | $780 \pm 4$ | $771 \pm 4$ | $400 \pm 2$ | $13 \pm 1$ | $11 \pm 1$ | $10 \pm 1$ |
| ground truth | SMC | Gaussian | $717 \pm 4$ | $715 \pm 4$ | $378 \pm 7$ | $12 \pm 1$ | $10 \pm 1$ | $11 \pm 1$ |
| pre-trained | CEM | Gaussian | $280 \pm 8$ | $108 \pm 12$ | $365 \pm 16$ | $9 \pm 1$ | $3 \pm 1$ | $19 \pm 2$ |
| pre-trained | SMC | Gaussian | $231 \pm 7$ | $244 \pm 8$ | $411 \pm 15$ | $8 \pm 1$ | $-7 \pm 1$ | $15 \pm 1$ |
| – | – | pre-trained | $615 \pm 7$ | $693 \pm 7$ | $56 \pm 1$ | $277 \pm 27$ | $318 \pm 35$ | $36 \pm 1$ |
| ground truth | CEM | pre-trained | $1291 \pm 33$ | $1218 \pm 35$ | * | $187 \pm 12$ | $275 \pm 18$ | * |
| ground truth | SMC | pre-trained | $1064 \pm 27$ | $1207 \pm 27$ | $373 \pm 9$ | $112 \pm 14$ | $472 \pm 23$ | $178 \pm 4$ |
| pre-trained | CEM | pre-trained | $1222 \pm 41$ | $1097 \pm 62$ | $349 \pm 16$ | $338 \pm 39$ | $462 \pm 50$ | $100 \pm 5$ |
| pre-trained | SMC | pre-trained | $1177 \pm 40$ | $1095 \pm 40$ | $420 \pm 14$ | $247 \pm 12$ | $473 \pm 17$ | $170 \pm 7$ |

Table 1: MPC with the true reward function, MuJoCo simulator/pre-trained dynamics model and fixed Gaussian/pre-trained proposals. We present best results from large sweeps over planner hyper parameters, compute budget, planner horizon and Gaussian proposal variance. Results use a horizon of 30 time-steps, 2000 samples and are averaged over 100 episodes. For further details, see Sec. F.1. * indicates settings where we saw numerical issues with ground truth model and the CEM planner.

parameter sweep are shown in Table 1, together with performance of the two proposals and baseline performance of a successful agent whose data was used to train the pre-trained model and proposal (labelled 'Near-Optimal performance'). Results from an additional sweep comparing performance of planning w.r.t computational budget and proposal use can be seen in Sec. F.1.

When planning with a zero-mean Gaussian proposal and the ground truth dynamics, both planners improve significantly on the performance of the Gaussian proposal for the simpler Ant tasks but not for the harder OP3 tasks (this may be in part because the OP3 tasks terminate if the robot falls, which makes planning hard). This result holds across a wide range of computational budgets (see subsection F.1). Planning with a pre-trained model and the Gaussian proposal performs similar to the proposal itself on the harder OP3 tasks but there is a small improvement, albeit less than when planning with the ground truth dynamics, for the simpler Ant tasks. Specifically on the Ant GTTP task, planning leads to better rewards but fails to consistently achieve target poses and there remains a large qualitative difference between the planner results and near-optimal performance.

When planning with the pre-trained task-agnostic proposal instead, the performance far exceeds the results from planning with the Gaussian proposal or executing the pre-trained proposal without any planning. This highlights the need for a suitable proposal, especially for the high-dimensional OP3 tasks, further motivating our approach learn a proposal for MPC with model-free RL. When planning with a pre-trained proposal, using the ground truth dynamics or a pre-trained model achieve similar performance, indicating that our pre-trained models are suitable for planning. Lastly, both planners perform similarly across tasks, though on the harder GTTP tasks SMC slightly outperforms CEM. We use SMC throughout the paper as it makes better use of the proposal unlike CEM which uses it only for plan initialization (see Sec. F.1 for full results).

## 5.2 Leveraging MPC with a learned model and proposal

Next, we evaluate different approaches to learning a proposal from scratch for MPC based on the approach discussed in Sec. 4.3. In particular, we consider the following variants:

- `MPO`: MPO with a distributional critic similar to Hoffman et al. (2020). Corresponds to $\alpha = 1, \beta = 0, p_{plan} = 0$ from Eqn. 4 and Alg. 1. `MPO` hyper-parameters are tuned for each task (and listed in Sec. F.2 of the supplement).
- `MPC+MPO`: MPC to collect data. MPO objective for learning ($\alpha = 1, \beta = 0, p_{plan} = 0.5$).
- `MPO+BC`: Adding MPO and BC objectives ($\alpha = 1, \beta > 0, p_{plan} = 0$), where $\beta$ is tuned per task.
- `MPC+MPO+BC`: MPC to collect data. Combined MPO+BC objective ($\alpha = 1, \beta > 0, p_{plan} = 0.5$).

The model is trained from scratch for the MPC variants (see Sec. 4.2). We choose $p_{plan} = 0.5$ for all our MPC experiments as it worked better than $p_{plan} = 1.0$, and use 250 samples and a planning horizon of 10 for SMC (see Sec. F.2 for ablations). We tune the BC objective weight $\beta$ per task.

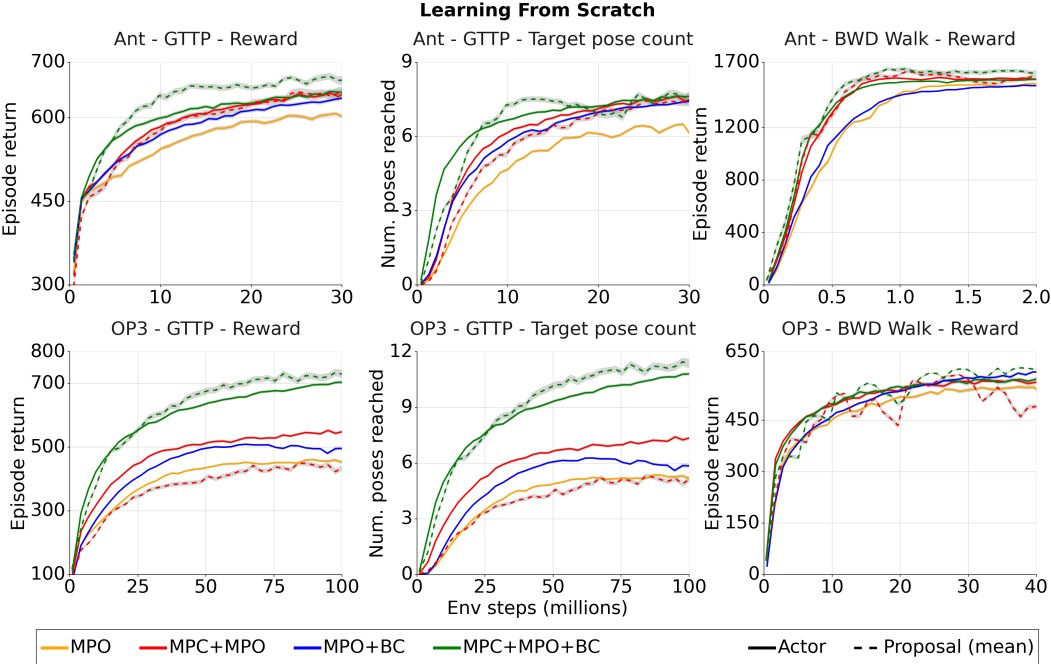

Figure 2: Performance of algorithmic variants in Sec. 5.2, when trained from scratch. **Actor**: Executing the MPC actor (`MPC+`) or the learned policy (rest). **Proposal (mean)**: Executing the learned proposal's mean action.

Figure 2 (left & center columns) presents performance comparisons on the go to target pose tasks[2], showing the reward and corresponding number of achieved target poses for each variant. `MPC+MPO` outperforms `MPO` significantly with regards to actor performance (see left column, solid lines). However even though `MPC+MPO` led to higher reward trajectories earlier in the learning process, the proposal distribution by itself tended to perform only as well as the `MPO` baseline. We speculate that this phenomenon is due to the increased off-policyness of the planner trajectories. Adding the BC objective mitigates this issue, improving the performance of both `MPO`[3] and `MPC+MPO`. The resulting variant `MPC+MPO+BC` significantly outperforms all other baselines in terms of actor and proposal performance as well as learning speed. Additionally, we tried bootstrapping with the learned critic for planning, but did not observe significant improvements on the GTTP tasks (see Fig. 12).

We also tested our approach on simpler forward and backward walking tasks for both the Ant and OP3 bodies. Figure 2 (right column) shows the results from the backward walking tasks. For the Ant (top row), `MPC+MPO` significantly outperforms `MPO` early on during training but reaches similar asymptotic performance while for the OP3 (bottom row) the difference between `MPO` and `MPC+MPO` is small throughout training. Interestingly, on these simpler tasks the proposal for `MPC+MPO` matches the actor performance and adding the BC objective gives only minor performance gains. We posit that a well-tuned implementation of the model-free `MPO` baseline achieves near-optimal performance on these tasks and provides a strong baseline that is hard to beat both in terms of data efficiency and performance even with the true reward function and bootstrapping with a learned critic. We present further details as well as forward walking results showing similar trends in the supplement (Fig. 13). Lastly, since the BC objective substantially improves results for both `MPO` and `MPC+MPO` we use the BC variants in all further experiments. Videos of learned behaviors can be seen at our website.

**Ablations:** We additionally ran several experiments ablating our design choices. The results of these experiments can be found in the supplement. Figure 10 and Figure 11 show the results of varying $p_{plan}$ as well as the number of samples used by the planner and the planning horizon, respectively. In Figure 12 we show the effect of bootstrapping with the learned value function within MPC. Figure 17 and Figure 18 compare deterministic models with PETS-style stochastic models (Chua et al., 2018)

---

[2]All learning curve results are averaged over 4 seeds. See Sec. F.2 for hyperparameter details and videos.

[3]Recent work has shown that similar combinations of `MPO` and a BC objective work quite well across challenging online and offline-RL settings (Abdolmaleki et al., 2021).

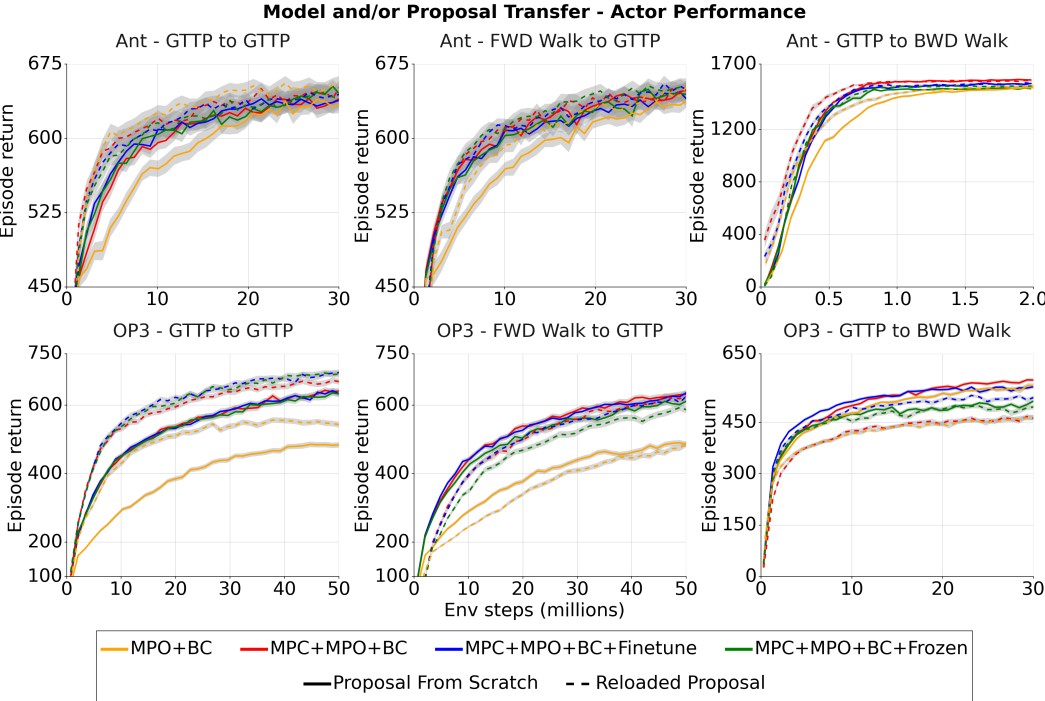

Figure 3: Performance of algorithmic variants in Sec 5.3 &5.4 with model and/or proposal transfer across tasks. Top / bottom row: Ant / OP3 results; Solid / dashed lines: Model transfer (except `MPO+BC`) / proposal transfer.

in our setting. Furthermore, in Figure 19 we report results using ensembles rather than a single model. Finally, Figure 20 shows results on the walker and humanoid tasks in the DeepMind Control Suite.

## 5.3 MODEL TRANSFER

In this section, we study how well models can transfer between tasks. We first explore to what extent a learned model can boost performance on a complex task. As a control experiment we transfer learned models from the OP3 GTTP task to the same task. We consider three different settings (in addition to the baseline `MPO+BC`) where a) no model is transferred and one is trained from scratch on the target task (`MPC+MPO+BC`), b) the transferred model is finetuned on the target task (`MPC+MPO+BC+Finetune`) and c) the transferred model is kept frozen throughout training on the target task (`MPC+MPO+BC+Frozen`). In this experiment (Figure 3 left column, solid lines) we find little improvements in learning speed or asymptotic performance when transferring a model vs learning from scratch. Transferring a frozen model (which can fail to generalize to out of distribution data) performs slightly worse vs finetuning the transferred model. This trend also holds in other transfer settings such as transferring models from the forward walking task to GTTP and transferring from the GTTP task to backward walking (Figure 3, center/right column, respectively, solid lines).

While this result is perhaps surprising, it does agree with our initial investigation of planning with learned models (Table 1) where we saw poor performance on all our tasks without a good proposal for the planner to leverage. Transferring a good model still does not solve the initial exploration problem, especially on the harder tasks. Videos of learned behaviors can be seen here.

## 5.4 PROPOSAL (AND MODEL) TRANSFER

In addition to transferring learned models across tasks we also consider transferring learned proposals. On each source task we train a proposal that is dependent only on proprioceptive information and lacks task-specific information (similar to the one used in subsection 5.1) and is thus transferable across tasks. As a simple approach to transfer a pre-trained proposal we used it for action generation while keeping the learning objective unchanged. Concretely, we use a mixture of the reloaded proposal from a source task and the learned proposal on the target task as our proposal distribution ($\pi_\theta$) for the MPC loop in algorithm 1. The mixture weight of the reloaded proposal is annealed linearly from 1 to 0 in a fixed number of learning steps (tuned per task); see Sec. F.3 of the supplementary material.

Again, we first transfer the proposal trained on the GTTP task to the same task (Figure 3, left column, dashed lines). Transferring the proposal leads to faster learning for both `MPO+BC` and `MPC+MPO+BC` as well as to a smaller extent, better asymptotic performance. Transferring the model and proposal together (`MPC+MPO+BC+Finetune`, `MPC+MPO+BC+Frozen`) does not lead to any additional improvements, further strengthening our intuitions from the model transfer experiments.

Next we transfer proposals between different source and target tasks, which yielded some nuanced insights into the need for compatibility of the tasks. Figure 3 (center column, dashed lines) shows the results when transferring a proposal from the forward walking task to the GTTP task and Figure 3 (right column, dashed lines) presents proposal transfer results from the GTTP task to backward walking. Interestingly, proposal transfer hurts both `MPC+MPO+BC` and `MPO+BC` in both these cases, especially for the high-dimensional OP3, leading to slower learning and lower final performance on both the target tasks. These results provide complementary insights regarding proposal transfer: there should be good overlap between the data distributions of the source and target tasks for proposal transfer to succeed. This is not the case in these transfer experiments; forward walking has a very narrow goal distribution compared to GTTP making the resulting proposal far too peaked, and while a proposal from the GTTP task would be quite broad (see website for some trajectory rollouts) it is highly unlikely to capture the behavior of walking backwards. Overall, these results provide encouragement that combining the right proposal, potentially trained on a diverse set of tasks, together with model-based planning can lead to efficient and performant learning on downstream tasks. For more transfer results and plots showing amortized policy performance see the supplement (Sec. F.3).

## 6    DISCUSSION

Our initial experiments highlighted that a good dynamics model is not enough to solve challenging locomotion tasks with model predictive control when computation time is limited, especially in tasks with high task and control complexity. In such settings a good proposal is necessary to guide the planner. Motivated by this finding we study different approaches to learning proposals for MPC, considering variants that combine the learning objective from MPO, a model-free RL algorithm, together with a behavioral cloning objective for efficient planner amortization. We also evaluate transfer performance across tasks where either the proposal, learned model, or both are transferred. Overall our results show that for the locomotion domains considered in this paper, MPC with a learned model and proposal can yield modest improvements in data efficiency relative to well-tuned model-free baselines. We found that the gains are larger as task and control complexity increase. On simpler walking tasks we saw very small improvements that are further diminished if an amortized policy is desired at test time. On our most challenging task, the OP3 go to target pose task, we see both significantly faster learning speed and improved asymptotic performance.

A common justification for model-based approaches is the intuition that models can more easily transfer to related tasks since the dynamics of a body are largely task independent. We attempted to validate this intuition but had difficulty achieving large gains in data efficiency even when transferring to the same task. We speculate that this finding is related to the difficulty of planning with a limited search budget in a multi-goal/multi-task setting even with a near perfect model. If the overall system is limited by the lack of a good proposal then model transfer by itself may have a negligible effect. When transferring models and proposals to new simpler tasks we also did not find substantial benefits. There are a number of other potential pitfalls in this setting in addition to the lack of a good proposal. If a proposal leads to trajectories that are inconsistent with the state distribution on which the model was trained, MPC may add little value. As we observed when transferring from the go-to-target-pose tasks to walking backwards, an unsuitable proposal may limit asymptotic performance. Additionally, on tasks with fairly narrow goal distributions a well-tuned model-free method can perform just as well as a model-based agent. This suggests tasks with multi-task/multi-goal settings provide a good test bed to showcase the strengths of model-based approaches and aid further research.

On the whole, the gains from MPC with learned models in our setting are meaningful, but not so dramatic as to be a silver bullet, a finding similar to Springenberg et al. (2020); Hamrick et al. (2021). This paper focused on learning complex locomotion tasks from state features, using a structured dynamics model as well as focusing on MPC as the way to leverage the model. There are a number of additional settings where models can be used differently and may aid transfer. For example, in partially observed tasks with pixel observations, transferring models and representations may lead to improvements in data efficiency (Byravan et al., 2020; Hafner et al., 2018; 2020).

ACKNOWLEDGMENTS

We would like to thank Jackie Kay, Alex X. Lee, Claudio Fantacci, Jessica Hamrick and Tom Erez as well as the rest of the team at Deepmind for insightful discussions and feedback. The data used in this project was obtained from mocap.cs.cmu.edu. The database was created with funding from NSF EIA-0196217.

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

# A  TASKS

We consider a number of locomotion tasks of varying complexity, simulated with the MuJoCo (Todorov et al., 2012) physics simulator. We consider two embodiments: an 8-DoF ant from `dm_control` (Tunyasuvunakool et al., 2020) and a model of the Robotis OP3 robot with 20 degrees of freedom. For each embodiment, we consider three tasks: walking forward, walking backward and "go-to-target-pose" (GTTP), a challenging task that is the focus of our evaluation. In all tasks, the agent receives egocentric proprioceptive observations (joint angles, joint, linear and angular velocities as well as end effector positions). In addition we provide the world z-axis in the robot's frame of reference. For the ant the proprioceptive observations are 37 dimensional. For the OP3 robot the proprioceptive observations are 64 dimensional. In addition to proprioceptive observations the agent also receives task specific observations that describe the task goal and differ by task.

## A.1  WALKING TASKS

In the walking forward and backward tasks the agent is rewarded for maximizing forward (or backward) velocity in the direction of a narrow corridor:

$$r = v \cdot e_{\text{target}}, \tag{5}$$

where $v$ denotes the velocity of the robot and $e_{\text{target}}$ denotes a unit vector in the direction of desired movement (both in the frame of the agent). For the OP3 robot we also include a small pose regularization term to regularize towards a walking pose. The agent observes the target direction $e_{\text{target}} \in \mathbb{R}^3$ in it's egocentric frame of reference as a task-specific observation for the walking tasks.

## A.2  GO-TO-TARGET-POSE TASK

The GTTP task consists of either body on a plane, with a target pose in relative coordinates as a task-specific observation and proximity to the target pose rewarded. When the agent is within a threshold distance of the target pose (i.e. it achieves the current target), it gets a sparse reward and a new target is sampled. For the ant we use target poses from policies trained on a standard go-to-target task (Tunyasuvunakool et al., 2020). For the OP3, we use poses from the CMU mocap database (cmu) (retargeted to the robot). We use thousands of different target poses; the agent has to learn to transition between

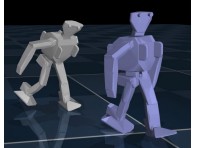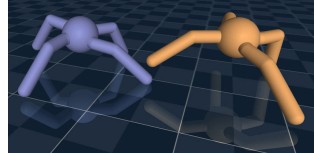

Figure 4: Go-to-target-pose (GTTP) task with the OP3 & Ant. The agent has to reach the target pose (blue).

them. Thus the GTTP task can be seen as either a single highly diverse task or as a multi-task setting with strongly related tasks and consistent dynamics. We extend existing motion tracking infrastructure (Hasenclever et al., 2020) to build our task.

The agent agent receives task specific observations that specify the target pose relative to the agent. Specifically, we use relative joint angles, as well as relative root position and relative positions and orientations of a number of different body parts, all expressed in the egocentric frame of the embodiment. These task observations are 107 dimensional for the ant and 163 dimensional for the OP3, respectively.

**Task reward** The task reward consists of two parts, a dense reward term that corresponds to how well the target pose is matched and a sparse reward that is added if the target pose is 'achieved':

$$r = r_{\text{dense}} + S\mathbb{1}(r_{\text{dense}} > r_{\text{threshold}} \text{ for N steps}), \tag{6}$$

where we use $r_{\text{threshold}} = 0.65, N = 1, S = 50$ for the OP3 and $r_{\text{threshold}} = 0.8, N = 10, S = 50$ for the ant. The rewards are similarly robot specific and will be described below. The rewards as well as the tolerance $r_{\text{threshold}}$ and $N$ were tuned to give visually good behaviors with a large-scale on-policy agent.

For the dense reward we use rewards of the following form:

$$r_{\text{dense}} = 0.6 \exp\left(-\left(\frac{m_{\text{robot}}d}{0.3}\right)^2\right) r_{\text{pose}} + 0.4 \exp\left(-\left(\frac{m_{\text{robot}}d}{2}\right)^2\right), \tag{7}$$

where $d$ denotes the center of mass distance between the agent and the target, $r_{\text{pose}}$ is a reward that depends chiefly on the robots joint configuration but less on the relative position to the target pose and $m_{\text{robot}}$ is a robot specific multiplier. We use $m_{\text{robot}} = 4$ for the OP3 and $m_{\text{robot}} = 1$ for the ant.

**Ant pose reward** For the ant we use the following pose reward:

$$r_{pose} = 0.5 \exp\left(-d_{\text{root}}\right)\left(0.4 r_{\text{pos}} + 0.6 r_{\text{quats}}\right) + 0.5 \exp\left(-0.1 d_{\text{root}}\right), \tag{8}$$

where $d_{\text{root}}$ denotes the relative root distance. $r_{\text{pos}}$ and $r_{\text{quat}}$ are terms penalizing deviations from the target pose in terms of the positions and orientations of the individual MuJoCo bodies that make up the Ant:

$$r_{\text{pos}} = \exp\left(-0.85\|p_{\text{bodies}} - p_{\text{bodies}}^{\text{ref}}\|^2\right) \tag{9}$$

$$r_{\text{quat}} = \exp\left(-4\|q_{\text{bodies}} \ominus q_{\text{bodies}}^{\text{ref}}\|^2\right), \tag{10}$$

where $p_{\text{bodies}}$ and $p_{\text{bodies}}^{\text{ref}}$ are the body positions of the ant and the reference, respectively, and $q_{\text{bodies}}$ and $p_{\text{bodies}}^{\text{ref}}$ are the body positions of the ant and the reference, respectively,

**OP3 pose reward** For the OP3 we use the following pose reward (similar to Peng et al. (2018); Hasenclever et al. (2020)) with terms penalizing deviations in terms of the center of mass, the joint angle velocities, the end effector positions and the joint orientations:

$$r_{\text{pose}} = 0.1 r_{\text{com}} + r_{vel} + 0.15 r_{\text{app}} + 0.65 r_{\text{quat}}$$

The first term $r_{\text{com}}$ penalizes deviations from the centre of mass:

$$r_{\text{com}} = \exp\left(-40\|p_{\text{com}} - p_{\text{com}}^{\text{ref}}\|^2\right), \tag{11}$$

where $p_{\text{com}}$ and $p_{\text{com}}^{\text{ref}}$ are the positions of the centre of mass of the simulated character and the mocap reference, respectively. The second term $r_{\text{vel}}$ penalizes deviations from the reference joint angle velocities:

$$r_{\text{vel}} = \exp\left(-0.1\|q_{\text{vel}} - q_{\text{vel}}^{\text{ref}}\|^2\right), \tag{12}$$

where $q_{\text{vel}}$ and $q_{\text{vel}}^{\text{ref}}$ are the joint angle velocities of the simulated character and the mocap reference, respectively. The third term $r_{\text{app}}$ penalizes deviations from the reference end effector positions:

$$r_{\text{app}} = \exp\left(-160\|p_{\text{app}} - p_{\text{app}}^{\text{ref}}\|^2\right), \tag{13}$$

where $p_{\text{app}}$ and $p_{\text{app}}^{\text{ref}}$ are the end effector positions of the agent and the target pose, respectively. Finally, $r_{\text{quat}}$ penalizes deviations from the reference in terms of the quaternions describing the body orientations:

$$r_{\text{bodies}} = \exp\left(-2\|q_{\text{quat}} \ominus q_{\text{bodies}}^{\text{ref}}\|^2\right), \tag{14}$$

where $\ominus$ denotes quaternion differences and $q_{\text{bodies}}$ and $q_{\text{bodies}}^{\text{ref}}$ are the joint quaternions of the agent and the target pose, respectively.

**Planner rewards** The reward terms above cannot be computed solely from observations available to the agent. In principle it would be possible to instead use reward terms that are only based on observations. However, in practice, we use planner rewards that are correlated with but different from the task reward.

For the OP3 we use the following reward for planning:

$$r = 0.5 \left(0.9 \exp\left(-d_{\text{root}}\right) + 0.1 \exp\left(-\frac{d_{\text{root}}}{10}\right)\right)\left(0.6 \exp\left(-\frac{2\|p_{\text{joints}} - p_{\text{joints}}^{\text{ref}}\|^2}{J}\right)\right)$$
$$+ 0.5 \exp\left(-\frac{d_{\text{root}}}{10}\right), \tag{15}$$

where $J$ is the number of joints.

For the Ant we use a similar reward with slightly different lengthscales for the planning reward:

$$r = 0.5 \left(0.9 \exp\left(-\frac{d_{\text{root}}}{4}\right) + 0.1 \exp\left(-\frac{d_{\text{root}}}{40}\right)\right)\left(0.6 \exp\left(-\frac{2\|p_{\text{joints}} - p_{\text{joints}}^{\text{ref}}\|^2}{J}\right)\right)$$
$$+ 0.5 \exp\left(-\frac{d_{\text{root}}}{40}\right), \tag{16}$$

**Target pose sampling** At the beginning of an episode and whenever a target pose is achieved we sample a new target. The target pose is sampled uniformly from a reference set of target poses. For the ant this set is derived from expert policies that have been trained on a standard go-to-target task from `dm_control` (Tunyasuvunakool et al., 2020). For the OP3 we use target poses from the CMU mocap database (cmu).

To determine the position of the new target relative to the agent we sample both a heading relative to current agent heading (sampled uniformly between $-120°$ and $120°$) and a random distance. We also sample the orientation of the target randomly relative to the current agent orientation (heading shift sampled uniformly between $-120°$ and $120°$).

To make this challenging task a bit easier we use what we call an *intra-episode curriculum* to determine the distance of the target pose from the agent. We linearly scale up the random distance with each achieved target (up to some maximum value). Concretely, let $U[a, b]$ be a uniform distribution and let $M$ be the number of curriculum phases. Then after having achieved $m$ target poses, the distance of the next target pose is drawn from $U[\min(m/M, 1)a, \min(m/M, 1)b]$. We use $M = 4$ and $a = 0.5, b = 5$ meters for the ant and $a = 0.05, b = 1.$ meters for the OP3.

**Motivating the go-to-target-pose task** The go-to-target-pose task can be motivated and understood from a few distinct perspectives. Firstly, it arises naturally as a temporally abstract version of a motion tracking task, and indeed we build our task as an extension of existing motion tracking infrastructure (Hasenclever et al., 2020) that is available as part of `dm_control` (Tunyasuvunakool et al., 2020). In a tracking task the target pose and reward change every timestep such that the agent is rewarded for producing behavior that tracks a time-varying reference motion. In the GTTP task, the instruction and reward definitions can be similar to the tracking task; however, the GTTP task involves a fixed target pose for a number of timesteps until the target is achieved. Insofar as the policy that solves a tracking task is an inverse dynamics model $(s, s' \rightarrow a)$, the policy that solves the GTTP task is a temporally abstract inverse model.

From another perspective, the GTTP task is the essential self-rearrangement task that can be performed by any avatar in an empty environment. Batra et al. (2020) argue that a conceptual unification of goal-directed navigation tasks, visual servoing, and object manipulation tasks are that they all are rearrangement tasks that can be standardized as challenges for embodied control.

Rearrangement tasks vary along multiple dimensions, including temporal abstraction and degree of specification. A motion tracking task is essentially a self-rearrangement challenge that involves a target pose being fully specified at every timestep. The GTTP task involves a target pose being specified with an unspecified offset and the policy must handle the temporal abstraction by closing in on the target pose. While both of these tasks fully specify the desired pose, one can alternatively consider versions of the tasks where the desired pose is not fully specified, or even further where the desired pose is about external objects rather than the pose of the body. To surmount the issues related to temporal abstraction, policies may, implicitly or explicitly, break problems down into subgoals. Curiously, goal-directed policies (or inverse models) also may themselves serve as effective abstractions for achieving subgoals, playing the role of lower-level controllers insofar as they can abstract away the capabilities required to execute movements to a subgoal.

We believe the GTTP task should be particularly amenable to model-based methods: it combines a high-dimensional control problem with a diverse goal distribution. This makes it hard to solve quickly with model-free methods. However, since the dynamics are shared between all goal poses, a dynamics models should be beneficial in leveraging the common structure.

## B  PLANNERS

We consider two sample based planners, primarily a Sequential Monte Carlo based non-iterative planner (SMC) (Piché et al., 2019; Gordon et al., 1993) and the Cross-Entropy Method (CEM) (Botev et al., 2013). Our SMC and CEM planners are shown in algorithm 2 and 3, respectively. We briefly describe the planners and discuss how they can be warm-started with a provided action proposal distribution.

**SMC planner:** The SMC planner (Alg. 2) is a non-iterative sample-based planner that maintains a number of particles $S$ corresponding to different model rollouts. All particles are initialized to

start at the initial state $s_0$. At each step of the rollout (upto a horizon of $H$), an action is sampled from a proposal for each particle; this proposal can be either a fixed distribution or the learned proposal. These actions are then rolled out through the model to compute the next state, reward and optionally, a value estimate. The particles are then resampled according to the weighted exponentiated reward/advantage with a temperature parameter $\tau$. The rollout is performed upto a horizon of $H$ and the first action from a randomly chosen particle (amongst the surviving particles at the final timestep) is chosen as the action to be executed. Unlike CEM, SMC samples actions from the proposal at each step within the rollout and is therefore more suited to warm-starting using a learned proposal; we choose it as our planner primarily for this reason.

---

**Algorithm 2** SMC planner

---

1: Given: state $s_0$, proposal $\pi_\theta$, model $m_\phi$, reward $r$, planning horizon $H$, number of samples $S$, planner temperature $\tau$ and optionally value function $V_\psi$

2: $\{s_0^{(i)} = s_0\}_{i=1}^S$          // *Initialise samples to the initial state*

3: Let $x_{0:t}^{(i)} = \{(s_0^{(i)}, a_0^{(i)}), \dots, (s_t^{(i)}, a_t^{(i)}), s_{t+1}^{(i)}\}$ denote the i$^{\text{th}}$ particle up to time $t$

4: **for** $t = 0 \dots H$ **do**

5:      $\{a_t^{(i)} \sim \pi_\theta(\cdot|s_t^{(i)})\}_{i=1}^S$        // *Sample actions*

6:      $\{s_{t+1}^{(i)} = m_\phi(s_t^{(i)}, a_t^{(i)})\}_{i=1}^S$        // *Take model step.*

7:      $\{r_t^{(i)} = r(s_t^{(i)}, a_t^{(i)}, s_{t+1}^{(i)})\}_{i=1}^S$        // *Evaluate rewards.*

8:      calculate advantage $A_t^{(i)}$ based on rewards and value function $V_\psi$
     // *Resampling*

9:      $w_t^{(i)} \propto \exp\left(A_t^{(i)}/\tau\right)$        // *Resampling weights proportional to exponentiated advantage*

10:     $i_1, i_2, \dots i_S \sim \text{Categorical}(\{w_t^{(i)}\}_{i=1}^S)$        // *Draw resampling indices*

11:     $x_{0:t}^{(k)} \leftarrow x_{0:t}^{(i_k)} \ \forall k = 1 \dots S$        // *resample trajectories*

12: **end for**

13: $i \sim \mathcal{U}\{1, \dots, S\}$        // *Sample action index*

14: **return** $a_0^{(i)}$        // *Return first action from a random particle*

---

**CEM planner:** The CEM planner (Alg. 3) is an iterative sample-based planner maintains a distribution over action sequences that is usually parameterized as a Gaussian with mean $\mu$ and standard deviation $\sigma$. At the start of planning, the mean $\mu$ is initialized to an open-loop sequence of actions sampled from a proposal distribution (either fixed or learned) and the standard deviation $\sigma$ is initialized to $\sigma_{init}$. In each iteration, $S$ trajectories are sampled from this distribution and rolled out through the model and their returns are computed. The top $E$ fraction of trajectories, ranked by the return, are retained and their mean $\mu_{elite}$ and standard deviation $\sigma_{elite}$ are computed. These are used to updated the mean $\mu$ and standard deviation $\sigma$ via an exponential average (with weights $\alpha_{mean}$ and $\alpha_{std}$). This procedure is repeated for $I$ iterations and after the final iteration, the first action from $\mu$ is executed in the environment. Unlike SMC, CEM uses the proposal only for plan initialization at the first iteration and uses the distribution $\mathcal{N}(\mu, \sigma)$ for further trajectory samples.

## C    Models

Successfully applying model predictive control requires learning good dynamics models. In this paper, we train single-step feedforward dynamics models $m_\phi$ that take in the current state $s_t$ and action $a_t$ and predict the next state $s_{t+1} = m_\phi(s_t, a_t)$. This dynamics model has both learned and hand-designed components; the task-agnostic proprioceptive observations are predicted via "black-box" neural networks while any non-proprioceptive – task-specific – observations (e.g. relative pose of the target) are calculated in closed form from the predicted proprioceptive observations in combination with a known kinematic model of the robot.

Fig. 5 (top) shows the network architecture of the learned components of the predictive dynamics model ($m_\phi$). We use a deterministic, feed-forward MLP per proprioceptive observation i.e. one each for joint angles, joint velocities, linear velocities and so on. This MLP takes as input all the proprioceptive observations and the action from the current timestep and predicts a delta change in the observation being predicted; this delta change is scaled by an embodiment specific timestep

---

**Algorithm 3** CEM planner

---

Given: state $s_0$, proposal $\pi_\theta$, model $m_\phi$, reward $r$, planning horizon $H$, number of samples $S$, elite fraction $E$, noise standard deviation $\sigma_{\text{init}}$, number of iterations $I$, and optionally value function $V_\psi$
// *Rollout proposal distribution using the model to initialize the plan.*
$(s_0, a_0, s_1, \ldots, s_H) \leftarrow \texttt{rollout\_with\_proposal}(m_\phi, \pi_\theta, H)$
$\mu \leftarrow [a_0, a_1, \ldots, a_H]$                    // *initial plan ($H \times$ action dimension)*
$\sigma \leftarrow \sigma_{\text{init}}$
**for** $i = 1 \ldots I$ **do**
    **for** $k = 1 \ldots S$ **do**
        $p_k \sim \mathcal{N}(\mu, \sigma)$                          // *Sample candidate actions.*
        // *Evaluate candidate action sequences open loop according to the model and compute associated returns.*
        $r_k \leftarrow \texttt{evaluate\_actions}(m_\phi, p_k, H, r, V_\psi)$
    **end for**
    Rank candidate sequences by reward and retain top $E$ fraction.
    Compute mean $\mu_{\text{elite}}$ and per-dim standard deviation $\sigma_{\text{elite}}$ based on the retained elite sequences.
    $\mu \leftarrow (1 - \alpha_{\text{mean}})\mu + \alpha_{\text{mean}}\mu_{\text{elite}}$            // *Update mean; $\alpha_{mean} = 0.9$*
    $\sigma \leftarrow (1 - \alpha_{\text{std}})\sigma + \alpha_{\text{std}}\sigma_{\text{elite}}$            // *Update standard deviation; $\alpha_{std} = 0.5$*
**end for**
**return** first action in $\mu$

---

($dt = 0.02$ for Ant, $dt = 0.03$ for OP3) and added to the observation at the current timestep to get the prediction at the next timestep:

$$\hat{s}_{t+1}^k = s_t^k + m_\phi^k(s_t, a_t) \cdot dt \tag{17}$$

where $\hat{s}_{t+1}^k$ denotes the k-th predicted observation at time $t + 1$, $s_t^k$ is the k-th observation at time $t$ and $m_\phi^k$ is the k-th MLP.

The only exception to this is for joint angle predictions, where we make a delta prediction on top of the joint velocity predictions to get the predicted joint angles at the next timestep:

$$\hat{q}_{t+1} = q_t + (m_\phi^q(s_t, a_t) + \hat{\dot{q}}_{t+1}) \cdot dt \tag{18}$$

Here, $q_t$ and $\hat{q}_{t+1}$ denote the observed and predicted joint angles at $t$ and $t + 1$ respectively, $m_\phi^q$ denotes the MLP that predicts the delta change in joint angles and $\hat{\dot{q}}_{t+1}$ is the predicted joint velocities at $t + 1$. In effect, we implement an Euler integration step for joint angles based on the predicted joint velocities $\hat{\dot{q}}_{t+1}$) and a correction term from the MLP $m_\phi^q(s_t, a_t)$.

In the case of the go-to-target-pose task, we have an additional learned "forward kinematics" model that predicts the 3D positions and orientations (represented as quaternions) of different parts of the embodiment. The architecture of this model is shown in Fig. 5 (bottom). This network takes in only the joint angles of the robot and predicts the 3D positions and orientations of the robot's bodies via a feed-forward, deterministic MLP; as before, there are two separate MLPs, one each for the position and orientation predictions.

## C.1 TRAINING

We train our models alongside the policy and critic using data collected from the MPC loop in Alg. 1 that is saved in the replay buffer. We sample a batch of trajectories $(s_{1:T}, a_{1:T-1}, r_{1:T})$ from this replay buffer, where $T$ is the trajectory length (set to 10 in all our experiments). This batch of trajectories is also used for policy and critic learning.

Given a trajectory of length $T$, we perform an open-loop rollout with the model using the initial state $s_1$ and the entire sequence of actions $a_{1:T-1}$:

$$\hat{s}_2 = m_\phi(s_1, a_1)$$
$$\hat{s}_3 = m_\phi(\hat{s}_2, a_2)$$
$$\ldots$$
$$\hat{s}_T = m_\phi(\hat{s}_{T-1}, a_{T-1}) \tag{19}$$

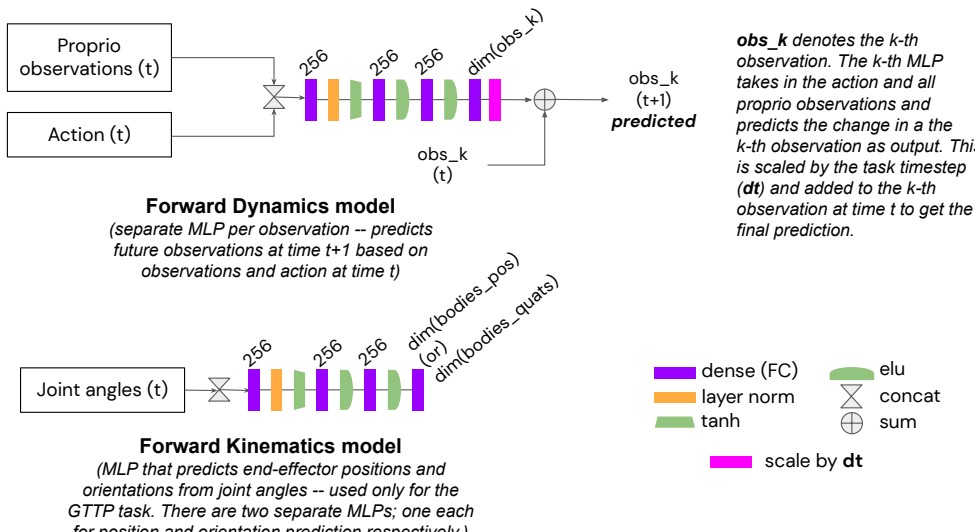

Figure 5: Network architectures of the forward dynamics and forward kinematics model. The forward dynamics model (top) takes as input the observations ($s_t$) and the actions ($a_t$) at time t, and predicts the observations at time t+1 ($s_{t+1}$). We use a separate MLP per predicted observation group that takes in the entire observation ($s$) as input and predicts a single observation group ($s^k$) as output (e.g. there is a separate MLP that regresses joint angles, one that regresses joint velocities etc.). This is done by predicting a delta change in the observation that is scaled by a task-specific timestep parameter ($dt = 0.02$ for Ant and $dt = 0.03$ for OP3 tasks) and added to the current observation $s_t^k$ to generate the final prediction ($\hat{s}_{t+1}^k$). We use separate MLPs for each observation group to decouple the different loss terms and handle scale differences between the different observation groups (e.g. accelerations and velocities tend to have larger values compared to positions). As the name suggests, the forward kinematics model (bottom) takes only the joint angles as input and predicts the positions and orientations of different parts of the embodiment. There is a separate network for position predictions and one for orientation predictions (represented as quaternions). This network is used only for the go-to-target-pose task where the task-specific observations include the relative pose of the target's bodies; we use the predictions of this network to integrate these reference observations in closed form (see Sec. C for an explanation).

where the first prediction $\hat{s}_2$ uses the true state $s_1$ and subsequent predictions use the predicted state from the previous timesteps.

We then compute a mean squared error between the predictions and the true states:

$$J_m = \frac{1}{T-1} \sum_{i=2}^{T} ||\hat{s}_i - s_i||^2 \tag{20}$$

where $\hat{s}_i$ and $s_i$ denote the predicted and true state respectively. This objective $J_m$ is minimized for training the model parameters $\phi$; we use an Adam optimizer (Kingma & Ba, 2014) with a fixed learning rate of 1e-4 for training, and use a sequence length of $T = 10$.

We use a similar training procedure for the forward kinematics model (used only for the GTTP task) but instead use the joint angles from the true states $s_{1:T}$ as inputs to the network as opposed to predictions. We compute a squared error between the true and predicted 3D positions and a quaternion difference for the between the true and predicted orientations; these are summed together and averaged across time to get the final objective for training. As before, we minimize this objective using an Adam optimizer (Kingma & Ba, 2014) with a fixed learning rate of 1e-4 and a sequence length of 10 for training the model parameters.

## C.2 INTEGRATING MODEL PREDICTIONS FOR PLANNING

As mentioned earlier, only proprioceptive observations are predicted by the learned models but the policy and critic depend on both proprioceptive and task-specific observations (see e.g. Fig. 6). Consequently, we need a way to generate future task-specific observations given an initial state and

action sequence for multi-step planning to work. We do this by integrating task-specific observations in closed form based on the predictions from the learned dynamics (and kinematics) models. As all our observations (proprio and task-specific) are specified in an egocentric frame of reference we do not need any external information for this integration other than the observations at the current timestep and the predictions from our learned models. We briefly describe the integration process for all our tasks next.

For the walking tasks, the only task-specific observation to be integrated is the relative direction of the target. Since the angular velocity of the robot is a proprioceptive observation (and the timestep $dt$ is known) we can generate a change in orientation by Euler integrating this velocity from the unit quaternion. By rotating the relative target direction with this change in orientation we can obtain the predicted relative target direction at $t + 1$.

For the GTTP tasks, there are several task-specific observations including the relative joint angle differences between the robot's current configuration and the target, the relative root position and the relative 3D positions and quaternions of the target bodies w.r.t to the current configuration. The integration of relative joint angles is straightforward: we can subtract the Euler integrated joint velocities to the relative joint angles at $t$ to get the relative joint angles at $t + 1$. Similarly, we can integrate the relative root position by adding the translation in the egocentric frame of reference (obtained by Euler integration of the linear velocities). The integration of the relative body positions and orientations is more involved, and uses a combination of the predicted translation, rotation (obtained by Euler integration of linear/angular velocities) and the observed & predicted positions and orientations of the robot (at $t$ and $t + 1$ respectively); as before, this does not need any external information not available as part of the egocentric observations. The procedure for predicting the full next state $\hat{s}_{t+1}$ given the current state $s_t$ and $a_t$ is as follows:

1. Predict all the proprioceptive observations via the learned dynamics model: $\hat{s}^k_{t+1} = m_\phi(s_t, a_t); \forall k$.

2. (Only for the GTTP task) Predict the body positions and orientations at $t+1$ via the predicted joint angles at $t + 1$ (from step 1 above).

3. Integrate the task-specific observations based on the predictions from the learned dynamics (and optionally, the kinematics) models.

This procedure is used within the PLANNER subroutine (see Alg. 1) for generating future states via multi-step rollouts.

## D MPO

Maximum A-Posteriori Policy Optimization (MPO) (Abdolmaleki et al., 2018b) is a continuous control RL algorithm that performs an expectation minimization form of policy iteration. There are two steps comprising **policy evaluation** and **policy improvement**. The *policy evaluation* step receives as input a policy $\pi_k$ and evaluates and action-value function $Q^{\pi_k}_\theta(s, a)$ by minimizing the squared TD error:

$$\min_\theta \mathbb{E}_{(s_t, a_t, s_{t+1}, a_{t+1}) \sim d^\pi} \left[ \left( r_t + \gamma Q^{\pi_k}_{\hat{\theta}}(s_{t+1}, a_{t+1}) - Q^{\pi_k}_\theta(s_t, a_t) \right)^2 \right],$$

where $\hat{\theta}$ denotes the parameters of a target network (Mnih et al., 2015) that are periodically updated from $\theta$ and $d^\pi$ denotes the distribution over $s_t, a_t, s_{t+1}, a_{t+1}$ induced by $\pi$. In practice we use a replay-buffer of samples in order to perform the policy evaluation step. The second step, *policy improvement* consists of optimizing the objective $\bar{J}(s, \pi) = \mathbb{E}_\pi [Q^{\pi_k}_\theta(s, a)]$ for states $s$ drawn from a state distribution $\mu(s)$. In practice the state distribution we draw samples from is the replay buffer. By improving $\bar{J}$ in all states $s$, we improve our objective. To do so, a two step procedure is performed.

First, we construct a non-parametric estimate $q$ such that $\bar{J}(s, q) \geq \bar{J}(s, \pi_k)$. This is done by maximizing $\bar{J}(s, q)$ while ensuring that the solution, locally, stays close to the current policy $\pi_k$; i.e. $\mathbb{E}_{\mu(s)}[\text{KL}(q(.|s)||\pi_k(.|s))] < \epsilon$. This optimization has a closed form solution given as

$$q(a|s) \propto \pi_k(a|s) \exp^{Q^{\pi_k}_\theta(s,a)/\eta},$$

where $\eta$ is a temperature parameter that can be computed by minimizing a convex dual function (Abdolmaleki et al., 2018a). Second, we project this non-parametric representation back onto a parameterized policy by solving the optimization problem $\pi_{k+1} = \arg\min_\pi \mathbb{E}_{\mu(s)}[\mathrm{KL}(q(a|s)||\pi(a|s)]$, where $\pi_{k+1}$ is the new and improved policy and where one typically employs additional regularization (Abdolmaleki et al., 2018b). Note that this amounts to supervised learning with samples drawn from $q(a|s)$; see Abdolmaleki et al. (2018b) for details.

### D.1 CHOICE OF MPO

We believe that the choice of the underlying learning algorithm (MPO) is not crucial for our results. In fact, we expect that using any state-of-the-art off-policy algorithm such as Soft Actor Critic (SAC) (Haarnoja et al., 2018) should lead to similar results across the tasks considered.

We choose MPO as our learning algorithm for two reasons. First, performance-wise, it is comparable with state-of-the-art off-policy actor-critic algorithms and has been thoroughly evaluated across a variety of continuous control settings (Hoffman et al., 2020). Second, the behavioral cloning objective (Eqn. 1) we add is compatible with the MPO objective (Eqn. 3) as they are both log-likelihood based objectives with a similar scale; setting a trade-off between the two is straightforward. Adding the BC loss to MPO has also been explored in prior work (Abdolmaleki et al., 2021) with improvements in learning across different settings.

## E  EXPERIMENTAL SETUP

We work with a distributed set-up with 64 actors interacting with the environment, and a single learner that trains the policy, critic and model. The actors collect data via the MPC loop described in Alg. 1, where the probability of planning $p_{plan}$ controls the tradeoff between executing the action from either the planner or the learned proposal $\pi_\theta$. This data is then stored in a replay buffer in the form of trajectories of state, action, reward triplets $(s_{1:T}, a_{1:T-1}, r_{1:T})$. The learner samples batches from the replay buffer (we use a batch size of 256, sequence length $T = 10$ and a replay buffer size of 1 million trajectories in all our experiments) and runs a learning update where the parameters of the policy, critic and model are updated in tandem. To study data efficiency in a data-limited regime, we additionally limit the number of actor steps per learner update step, i.e. the learner is blocked if the actor has not sampled enough data and the actor is blocked if the learner has not performed enough parameter updates. This is implemented using the reverb framework (Cassirer et al., 2021). We did a sweep over this "rate limit" parameter and found the best settings to be 8 actor steps per parameter update for the walking tasks and 16 actor steps per parameter update for the GTTP tasks (increasing the values further led to significantly reduced performance). This is held fixed for all from scratch and transfer experiments with all variants.

Alg. 1 describes the overall agent including the acting and learning loops which run in a distributed fashion with 64 actors and a single learner as mentioned earlier. The data from the actors which run MPC, warm-started with the learned policy (see MPC loop of Alg. 1) is added to the replay buffer; the learner asynchronously samples batches of data from this buffer and updates the parameters of the policy, critic, model and optionally, the task-agnostic proposal in an iterative fashion.

### E.1 POLICY AND CRITIC TRAINING

Our policy architecture is presented in Fig. 6 (top left, (a)); the policy takes as input both the proprioceptive and task-specific observations and outputs the parameters of a Gaussian distribution from which actions are sampled. We use a distributional critic (Fig. 6 - bottom left, (b)) that takes as input both the observations and actions and returns a discrete distribution (101 bins) that parameterizes a histogram of Q-values (from 0-300), similar to the one proposed in Hoffman et al. (2020).

Given a batch sampled from the replay buffer we compute Q targets using Retrace (Munos et al., 2016) with a terminal value bootstrapped from a target critic network (both the target policy and critic network parameters are updated once every 200 steps). The use of a distributional critic makes value backups hard; we circumvent this by bootstrapping using the mean Q-value. We then convert the value targets into a two-hot representation (only two bins in the histogram are non-zero, rest are zero), which we fit with our critic by minimizing the softmax cross-entropy loss between the critic

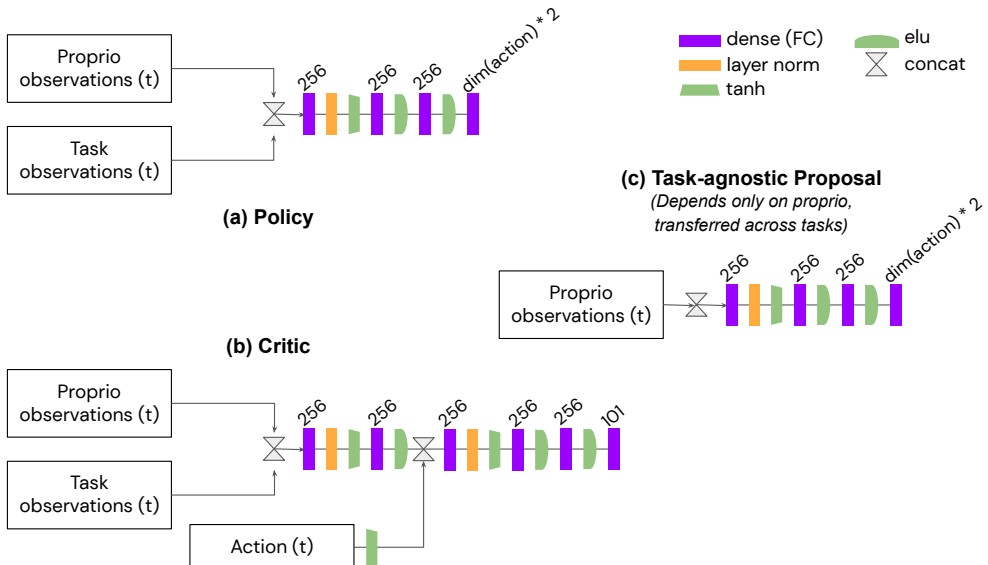

Figure 6: Network architectures of the policy, critic and task-agnostic proposal used for transfer. (a) The policy network takes all observations as outputs and predicts the mean/std. dev. of the Gaussian action distribution. The mean is scaled by a tanh and the standard deviation which is scaled via a softplus and summed together with an initial scale (1e-4 in our experiments). (b) The distributional critic takes all observations and the action at time t as input and predicts a discrete distribution that is made up of 101 bins (representing equally spaced Q-values from 0-300), and (c) The task-agnostic proposal only takes in the proprioceptive observations as input and predicts the parameters of a Gaussian distribution (similar to the policy network). Since this network does not take task-specific observations it captures only the average behavior, and can be freely transferred across tasks. We use this proposal for the initial planner experiments (Sec. 5.1) and the proposal transfer experiments (Sec. 5.4)

predictions and the two-hot targets. As discussed in the main text (Sec. 4.3 and Eqn. 4), we use a weighted combination of the model-free `MPO` objective and a Behavioral Cloning objective (BC) for training the policy and critic. We use separate Adam optimizers (Kingma & Ba, 2014) with learning rates of 3e-4 for training the policy and critic, and use an Adam optimizer with a learning rate of 1e-4 to optimize the temperature and dual variables in the MPO algorithm (Abdolmaleki et al., 2018b).

### E.2 TRAINING A TASK-AGNOSTIC PROPOSAL

In addition to the policy and critic, we also train a task-agnostic proposal (Fig. 6 - center right, (c)) that takes only proprioceptive observations as input and returns a Gaussian distribution for sampling actions. Since this proposal lacks any task-specific information it can only capture average behavior (e.g. walking randomly in the GTTP task, as opposed to directed walking to a target). Since proprioceptive observations are the same across tasks for a given embodiment this proposal can be transferred freely across tasks; we used this proposal for the initial planner experiments (Sec. 5.1) and the proposal transfer experiments (Sec. 5.4).

We train this task-agnostic proposal alongside our policy and critic using a Behavioral Cloning (BC) objective only. We sample batches from the replay buffer (which have both good and bad data) and fit the policy by maximizing the log-prob of executed actions under the proposal distribution (similar to Eqn. 1). We use an Adam optimizer with a learning rate of 1e-4 for training this task-agnostic proposal. We present a few videos showing the behavior learned by the task-agnostic proposals in the supplementary material and the website.

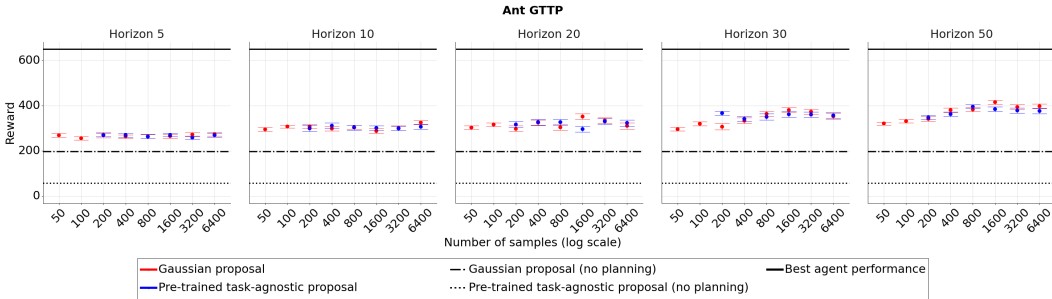

Figure 7: Performance of planning with SMC on the Ant GTTP task using a temperature of 0.01. Performance with both a standard Gaussian proposal and a pre-trained task-agnostic proposal improves when more samples are used. Interestingly, they perform similarly across the board. This is likely because the pre-trained proposal (which can walk but is not aware of the target pose) tends to propose actions that would wander away from the goal whereas the Gaussian proposal gives jittery actions that will not lead to sustained movement of the ant's centre of mass. Since the initial target pose is quite close to the ant and the ant is stable even jittery movements can yield a relatively large return. This is reflected in the performance of the two proposals without planning. The Gaussian proposal, which does not lead to much movement, achieves a larger return than the pre-trained proposal which wanders off. See our website for a visualization of the behavior generated by the task-agnostic proposal. 'Best agent performance' denotes the performance of the best RL agent from this work and should be interpreted as very good performance. Even with a large computational budget, planning does significantly worse than this agent. Results are averaged over 100 episodes. The error bars show $\pm$ 1 standard error.

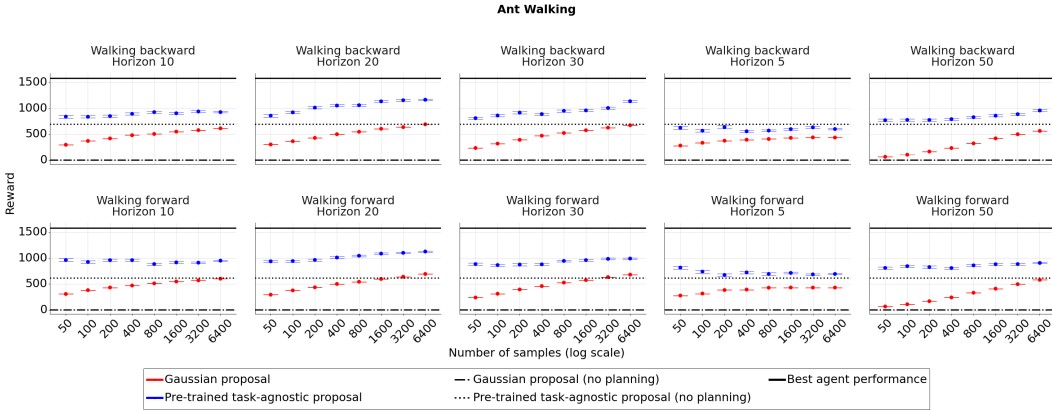

Figure 8: Performance of planning with SMC on the Ant walking tasks using a temperature of 0.1. Both a standard Gaussian proposal and a pre-trained task-agnostic proposal improve when more samples are used. As expected, the pre-trained proposal outperforms the Gaussian proposal across the board. Unlike the GTTP task the pre-trained task-agnostic proposal is able to walk approximately forward or backward but can drift. Thus, planning with this proposal does much better than using the Gaussian proposal (though the difference is reduced for larger computational budgets as one would expect). 'Best agent performance' denotes the performance of the best RL agent trained on this task and should be interpreted as very good performance. Results are averaged over 100 episodes. The error bars show $\pm$ 1 standard error.

# F  FURTHER EXPERIMENTAL RESULTS

## F.1  PLANNER RESULTS

For our result with pre-trained models and proposal (Table 1) we ran a hyper parameter sweep over the common planner parameters:

- Planning horizon ($H$): 5, 10, 15, 20, 25, 30
- Number of samples ($S$): 50, 100, 200, 500, 1000, 2000

For the Gaussian proposal we also swept over the standard deviation of the Gaussian (0.5, 1.0 and 2.0). In addition we also swept over planner-specific parameters. For SMC:

- Planner temperature ($\tau$): 0.001, 0.01, 0.1

For CEM:

- Elite fraction ($E$): 0.15, 0.3
- Number of iterations ($I$): 1, 2, 4
- Noise standard deviation ($\sigma_{init}$): 0.1, 0.25, 0.5, 1.0

Table 1 presents the best results for each planner, which use a planning horizon of $H = 30$ and number of samples $S = 2000$. We note here that best performance with CEM is obtained with $I > 1$, which effectively means that CEM uses a significantly larger computational budget than our SMC planner which is non-iterative; in spite of this SMC is still quite competitive with CEM across all tasks as can be seen from the results in Table 1.

In a further set of experiments we evaluated the performance of SMC with a wider range of samples and ground truth dynamics on the Ant walking, Ant GTTP and OP3 GTTP tasks. The results are shown in Figures 7, 8, 9. In the simpler, lower dimensional Ant domains, increasing the number of samples used by the planner improves performance, irrespective of the proposal distribution used for planning. In the higher-dimensional OP3 GTTP task the results are qualitatively different. When using longer horizons and more samples from a pre-trained, but goal-agnostic, proposal the performance increases. However even with 25600 samples from a standard Gaussian proposal the planner fails to make any progress. This is due to the difficulty of sample-based optimization in high-dimensional spaces, which further validates our choice of the use of a learned proposal distribution for the planner. Overall the planner results show the challenges of planning in challenging high dimensional tasks even with access to ground truth models, rewards and large computational budgets. Using a proposal is crucial for progress on these domains.

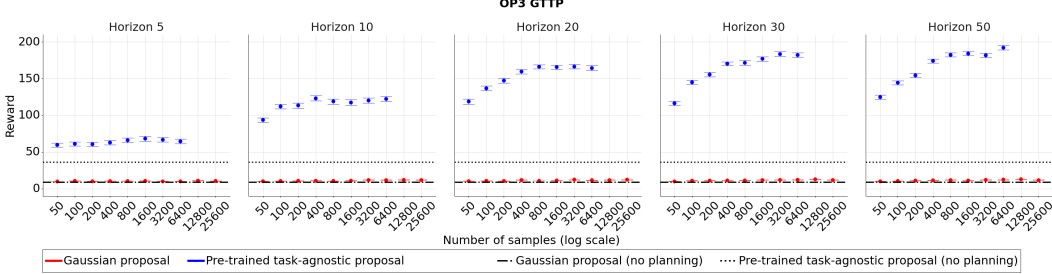

Figure 9: Performance of planning with SMC on the OP3 GTTP task using a temperature of $0.01$. With the pre-trained task-agnostic proposal performance improves when more samples are used. As with the Ant GTTP task this proposal can avoid falling and walk in random directions which provides significant reward compared to the Gaussian proposal which can easily cause the robot to fall, a state from which it cannot escape. Consequently, with a standard Gaussian proposal even 25600 samples do not lead to any meaningful task progress; on such high-dimensional tasks random sampling would rarely lead to any interesting behavior appearing. With the standard Gaussian proposal the planner reaches a target pose only in 1-2% of episodes by chance. The performance of the best RL agent trained on this task is around 730 (not shown) which is far better than anything achieved by planning (with or without the proposal). Results are averaged over 100 episodes. The error bars show $\pm 1$ standard error.

While the best planner only results use a high computational budget as presented above, we use SMC with $S = 250$ samples and planning horizon 10 for all our learning experiments for improved learning speed (wall-clock time) and to ensure that our computational budgets are realistic with respect to real-world considerations (assuming a 20Hz run-rate of the actor). Interestingly, this setting does not result in performance reduction compared to those with more compute as we show in ablation experiments (Fig. 10 and 11). Details on other planner hyper-parameters for individual experiments are detailed below.

## F.2 FROM SCRATCH RESULTS

In initial experiments we determined the number of actor steps per learner step (using the MPO baseline). We tried 4, 8 and 16 actor steps per learner step and found that the performance degraded

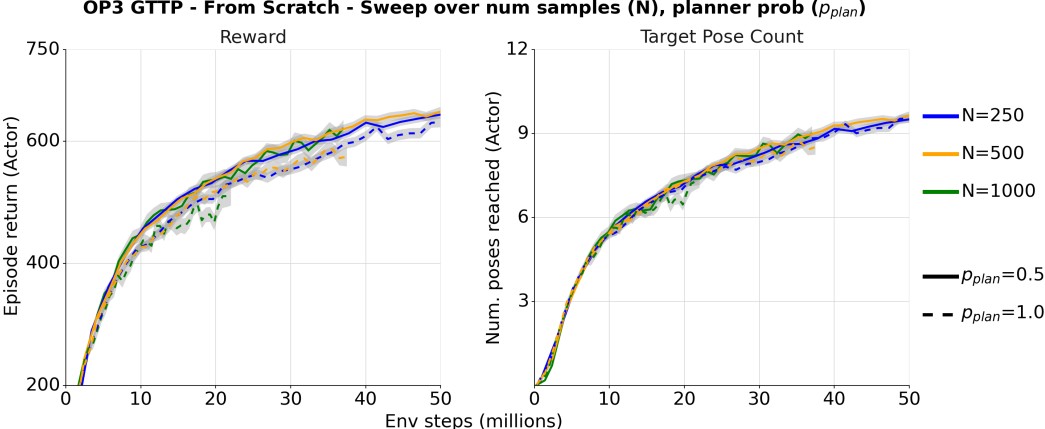

Figure 10: Performance of `MPC+MPO+BC` when learning from scratch on the OP3 GTTP task with a sweep over different number of samples for the SMC planner ($N$) and probability of planning ($p_{plan}$ from MPC loop of Alg. 1). Left: Plot comparing reward vs number of environment interactions, Right: Plot comparing the number of target poses reached vs number of environment interactions. For a fixed planning horizon $H = 10$, increasing the number of samples does not make a significant difference in performance (compared to the default $N = 250$, solid blue line). Increasing the probability of planning to $p_{plan} = 1.0$ (always planning) leads to slightly worse performance compared to interleaved planner and policy executions ($p_{plan} = 0.5$). Results averaged over two seeds.

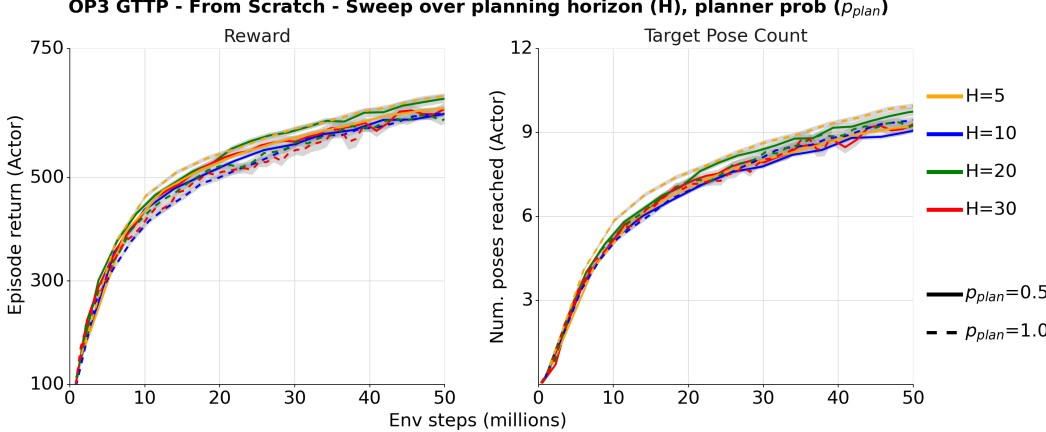

Figure 11: Performance of `MPC+MPO+BC` when learning from scratch on the OP3 GTTP task with a sweep over different planning horizons for the SMC planner ($H$) and probability of planning ($p_{plan}$ from MPC loop in Alg. 1). Left: Plot comparing reward vs number of environment interactions, Right: Plot comparing the number of target poses reached vs number of environment interactions. For a fixed number of samples $N = 250$, changing the planning horizon also does not have a significant impact on performance or data efficiency compared to the default setting of $H = 10$ (solid blue line); there is a slight improvement with $H = 20$ but this leads to significantly slower experiments w.r.t wall clock time so we choose $H = 10$ in our experiments. At lower horizons $H = 5$, always planning ($p_{plan} = 1.0$) does as well as interleaved planning and policy execution ($p_{plan} = 0.5$) but there's not a significant difference at higher horizons $H \geq 10$ (see left plot). Results averaged over two seeds.

when using lower than 8 actor steps per learner step for the walking tasks and 16 actor steps per learner step for the GTTP experiments. We used these settings for all subsequent experiments.

We tuned each algorithmic variant independently per task, running sweeps over MPO hyper parameters and, where applicable, BC objective weight $\beta$ and planner temperature $\tau$. We show the best results of each variant in the figures. For MPO hyper-parameters we ran sweeps over the constraint parameter $\epsilon = 0.1, 0.5$ in Equation 2. Additionally we sweep over several different settings for the additional trust-region constraint in the policy mentioned to in the discussion of Equation 3. In practice, MPO constraints the mean and variance of the policy separately, with constraint parameters

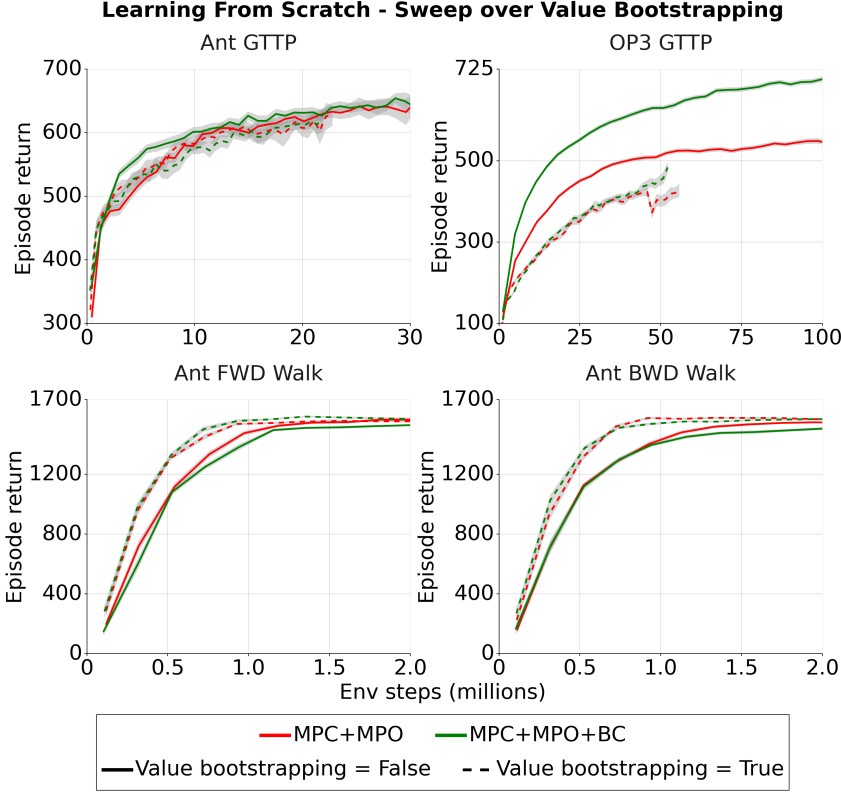

Figure 12: Performance of `MPC+MPO` and `MPC+MPO+BC` when learning from scratch on different Ant/OP3 tasks, when running a sweep over bootstrapping with the value function for planning (Alg. 1). Top row: Performance on the GTTP tasks with and without value bootstrapping (solid and dashed lines respectively). Learning a value function can be hard for the multi-task/multi-goal style GTTP task, and consequently bootstrapping with the value function does not improve performance, and in fact, hurts performance on the harder OP3 GTTP task (compare dashed and solid lines). Bottom row: Performance on the simpler Ant walking tasks, where value bootstrapping does improve performance and data efficiency. Results are averaged over four seeds.

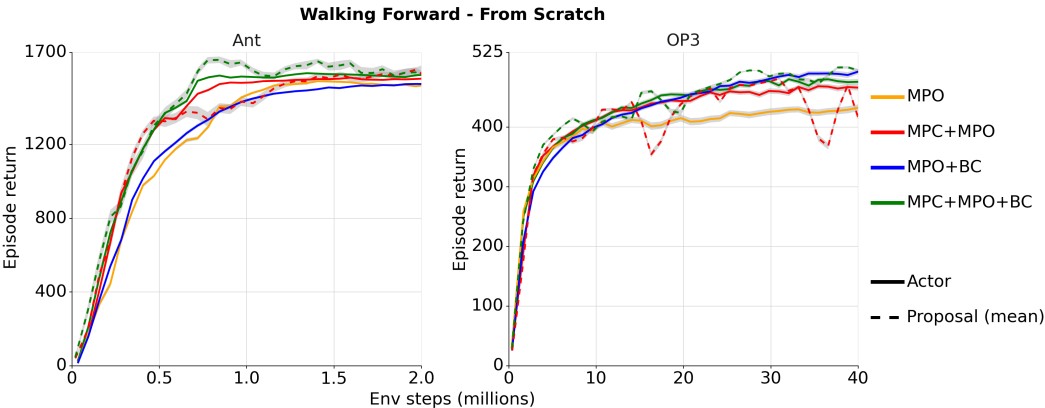

Figure 13: Performance of the various algorithmic variants in Sec 5.2 when trained from scratch on the Forward Walking task. Left column: Ant, Right column: OP3 results. `MPC+MPO` improves upon `MPO` for both the Ant and OP3 on the actor (solid lines) but the amortized policy has lower performance (red, dotted lines) than the actor. `MPO+BC` and `MPC+MPO+BC` improve on the performance of their non-BC counterparts, and the amortized policy of the `MPC+MPO+BC` matches or surpasses the corresponding actor performance (especially on the OP3 task). See Sec. 5.2 and Fig. 2 for a further description of the results when learning from scratch.

$\epsilon_\mu$ and $\epsilon_\Sigma$. This helps avoid premature convergence in some cases. See Abdolmaleki et al. (2018b) for details. We used $\epsilon_\Sigma = 10^{-5}$ and swept over 3 values from $\epsilon_\mu = 5 \cdot 10^{-4}, 10^{-4}, 5 \cdot 10^{-3}, 10^{-2}$

depending on the task. For algorithmic variants involving BC we ran hyper-parameter sweeps varying the BC objective weight $\beta = 0.001, 0.01, 0.1, 1.0$. For algorithmic variants involving MPC we ran sweeps over the planner temperature $\tau = 0.001, 0.01, 0.1$ ($\tau = 0.01$ worked best for all tasks except the Ant walking tasks where $\tau = 0.1$ worked best).

We use SMC with $S = 250$ samples and planning horizon of $H = 10$ for all our experiments with MPC. We determined these to be the best planner settings based on a hyperparameter sweep over planner parameters when learning from scratch with MPC on the OP3 GTTP task; these were also chosen with real-world considerations to ensure that planning could be run at 20Hz on practical systems. Fig. 10 presents the results when training `MPC+MPO+BC` with different number of samples $S$ for the SMC planner with a fixed horizon of 10, and the best temperature setting ($\tau = 0.01$) from the from scratch experiment (see Fig. 2). Increasing the number of samples or setting $p_{plan} = 1.0$ does not make a big difference. Fig. 11 shows a related experiment, but with a sweep over the planning horizon $H$ instead with $S = 250$ samples. Once again, there is little variation in performance with a longer or shorter horizon.

We also did a quick sweep over bootstrapping with the value function during planning, training `MPC+MPO+BC` from scratch on several tasks with and without bootstrapping. When bootstrapping from the value function, we use the learned critic and average the Q values from the given state and 10 actions sampled from the policy at the given state to compute the value $V$. Fig. 12 presents this ablation. In general, boostrapping with the value function helps on easier tasks such as walking (Fig. 12, bottom row) but hurts performance on the harder GTTP task (Fig. 12, top row) where learning a good value function can be difficult. We use the best settings for each task in our experiments (no bootstrapping for the GTTP tasks and bootstrapping for the walking tasks).

We additionally present results for learning from scratch on the walking forward tasks in Fig. 13. As with the backward walking results, `MPC+MPO` does better than `MPO` early on during learning and the addition of the BC objective improves performance compared to the non-BC counterpart especially on the OP3 task. All variants converge to similar asymptotic performance.

### F.3 TRANSFER RESULTS

**Model transfer:** As discussed in the main text we test two variants of model transfer where the transferred model is kept frozen on the target task (`MPC+MPO+BC+Frozen`), or finetuned on the target task (`MPC+MPO+BC+Finetune`), along with a baseline where we learn the model from scratch. For all the model transfer experiments we transfer the models from the best performing `MPC+MPO+BC` agent on the source task (and its hyperparameter settings). For all transfer settings except GTTP to GTTP we transfer only the forward dynamics model; for GTTP to GTTP we transfer both the forward dynamics and kinematics models. For finetuning we use an Adam optimizer with a learning rate of 1e-4 for both models (similar to from scratch training).

**Proposal transfer:** In the proposal transfer experiments we transfer the task-agnostic proposal which depends only on proprioceptive observations to the target task. We use a mixture of this reloaded proposal from the source task and the learned amortized policy (which has access to all observations) on the target task as the proposal distribution ($\pi_\theta$) for planning in algorithm 1. The mixture weight of this reloaded proposal is annealed linearly from 1 to 0 in a fixed number of learning steps $M$. At the start of learning on the target task we sample exclusively from the reloaded task-agnostic proposal on the actor and as learning progresses we sample less and less from this proposal; when the mixture weight reaches 0 we stop sampling from the reloaded proposal and revert to using only the amortized policy on the target task as the proposal distribution on the actor.

We tuned the hyper-parameter $M$ (in terms of learner steps) which controls the slope of this annealing of the mixture probability on all our transfer tasks and presented results from the best setting of $M$. We choose all other parameters based on the best from scratch results on the target task, with the BC objective added (`MPC+MPO+BC`).

The per-task sweeps are:

- OP3 GTTP to GTTP: $M$ = 1.25e5, 2.5e5, 5e5, 1e6, 2e6
- Ant GTTP to GTTP: $M$ = 15625, 31250, 62500, 125000, 250000
- OP3 Forward Walk to GTTP: $M$ = 1.25e5, 2.5e5, 5e5, 1e6, 2e6

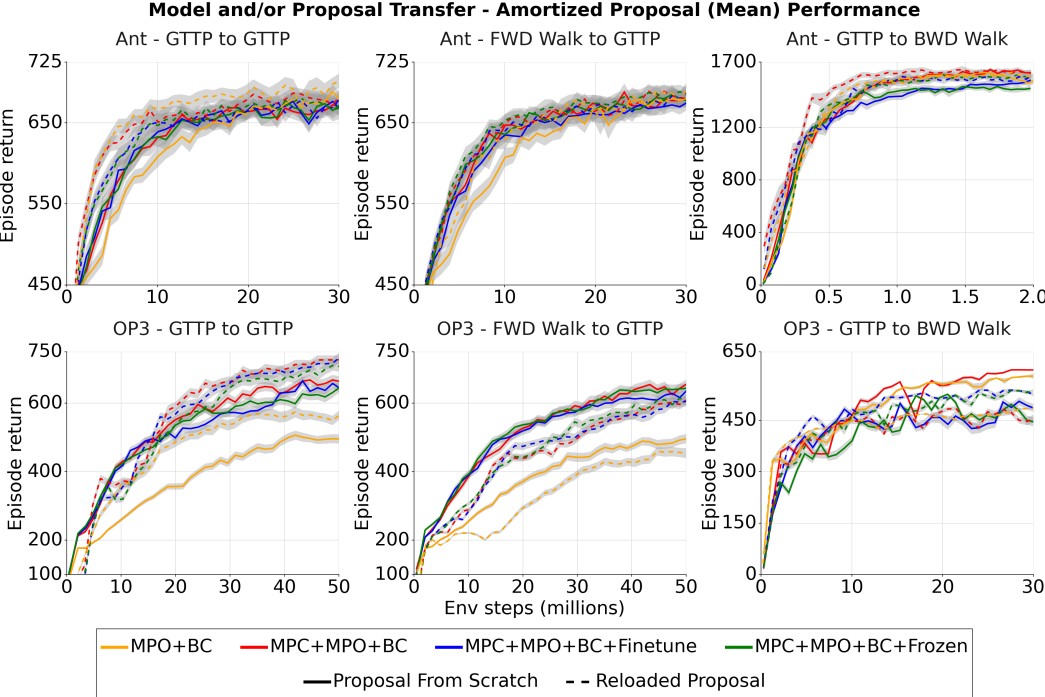

Figure 14: Performance of the amortized proposal (mean) for all the algorithmic variants in Sec 5.3 &5.4 with model and/or proposal transfer across tasks. Top / bottom row: Ant / OP3 results; Solid / dashed lines: Model transfer (except `MPO+BC`) / proposal transfer. The amortized policies converge to similar (or slightly higher) performance compared to the actor performance results shown in Fig. 3. As explained in Sec. 5.3, model transfer leads to fairly small performance improvements across tasks. Proposal transfer (Sec. 5.4) helps for transfer from the GTTP task to the GTTP task itself (left column) but can hurt performance when there is a mismatch in state/goal distributions between the source and target tasks, especially for the higher-dimensional OP3 (Forward walking to GTTP - center column & GTTP to Backward walking - right column).

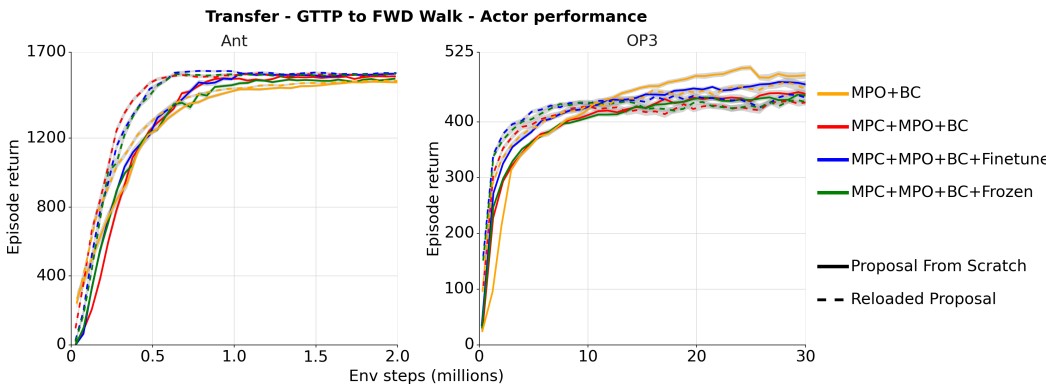

Figure 15: Actor performance for different model and proposal transfer variants from Sec. 5.3 and Sec. 5.4 when transferring models and/or proposals from the GTTP task to the Forward walking task. For the simpler Ant (left column), transferring the model does not lead to performance improvements but transferring the proposal does speed up learning significantly. For the OP3 (right column), transferring the proposals or models leads to faster learning at the beginning but results in convergence to a sub-optimal policy with lower performance compared to the baseline `MPO`.

- Ant Forward Walk to GTTP: $M = 15625, 31250, 62500, 125000, 250000$

- OP3 GTTP to Forward/Backward Walk: $M = 15625, 31250, 62500, 125000$

- Ant GTTP to Forward/Backward Walk: $M = 5000, 10000, 20000, 40000$

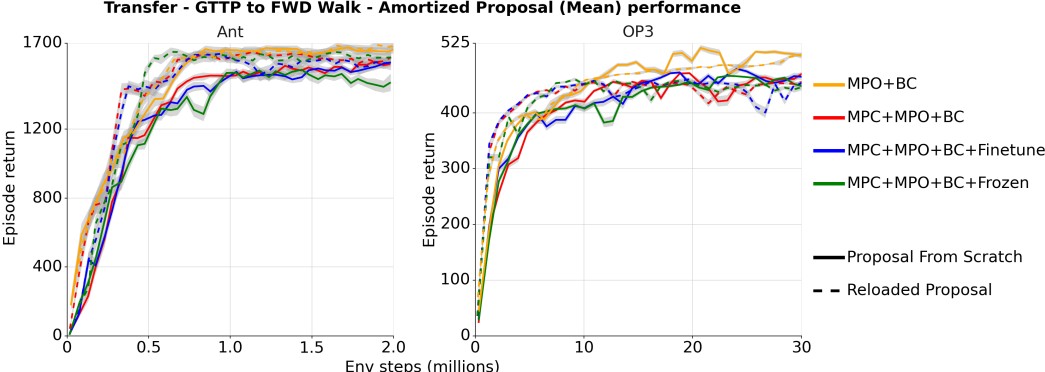

Figure 16: Performance of the amortized policy (mean) when transferring models and/or proposals from the GTTP task to the Forward walking task. Amortized policy matches (or slightly outperforms) the performance of the planner both during learning and asymptotically (see Fig. 15 to compare against actor performance).

**Additional transfer results:** Fig. 14 shows the performance of the learned amortized policy for the different transfer variants on several of our transfer tasks. This is the counterpart to Fig. 3 which shows the actor performance on the same tasks; in general the amortized policy is able to match or exceed the performance of the actor for all these settings due to the addition of the BC objective. Finally, Fig. 15 and Fig. 16 show the performance of the different variants when models and/or proposals are trained on the GTTP task and transferred to the Forward Walking task. As in previous transfer results (Sec. 5.3), model transfer (solid lines) does not lead to significant improvements w.r.t learning speed or asymptotic performance. Similar to the results in Sec. 5.4, proposal transfer (dotted lines) can help speed up learning early on but can also potentially hurt asymptotic performance especially on the more complex OP3 tasks.

### F.4 MODEL ABLATION - USING A STOCHASTIC MODEL ENSEMBLE

In our work we primarily used deterministic dynamics models (Sec. C), which is in contrast with some of the recent literature on model-based reinforcement learning that advocates for ensembles of stochastic models (e.g. PETS (Chua et al., 2018)). In this section we ablate this design choice. In this ablation, we use the same architecture as for our deterministic models but with Gaussian output distributions (parameterized by a mean and log-variance) instead of single predictions. Similar to PETS (Chua et al., 2018), we bound the predicted variance to ensure stability and use a single step negative log-likelihood loss for training the stochastic models instead of a multi-step loss as for the deterministic models.

For ensembles, we initialize each member randomly and train each ensemble member on a randomly selected subset (of half the batchsize) of each batch to encourage diversity of model predictions. For planning with the stochastic ensemble, we split the different particles in SMC equally across the ensemble members; each particle uses the same ensemble member throughout entire rollout (similar to the $\mathtt{TS}\infty$ approach from PETS (Chua et al., 2018)). This approach ensures that we keep the overall computational budget for planning the same irrespective of the ensemble size.

The results of this ablation study are presented in Fig. 17 and Fig. 18 where we compare the performance of $\mathtt{MPC+MPO}$ and $\mathtt{MPC+MPO+BC}$ when learning from scratch on all our tasks. Fig. 17 shows the performance of the actor; using ensembles of stochastic models does not lead to a significant difference in performance on the simpler Ant tasks and provides a small advantage on the OP3 tasks. This advantage is reduced when looking at the performance of the amortized proposal in Fig. 18; only the performance of $\mathtt{MPC+MPO}$ on the OP3 GTTP task is slightly improved through the use of the stochastic ensemble. All results used an ensemble size of 3 except the results on the Ant GTTP task which uses a single stochastic model.

We additionally ran an ablation on the size of the stochastic ensemble, considering ensembles of 1,3 and 5 models. Fig. 19 shows both the actor and amortized proposal performance of $\mathtt{MPC+MPO}$ and $\mathtt{MPC+MPO+BC}$ on the OP3 GTTP task. We observed no meaningful difference between 1, 3 and 5 ensemble members with the same computational budget for planning; similar results were observed for all other tasks considered in this paper.

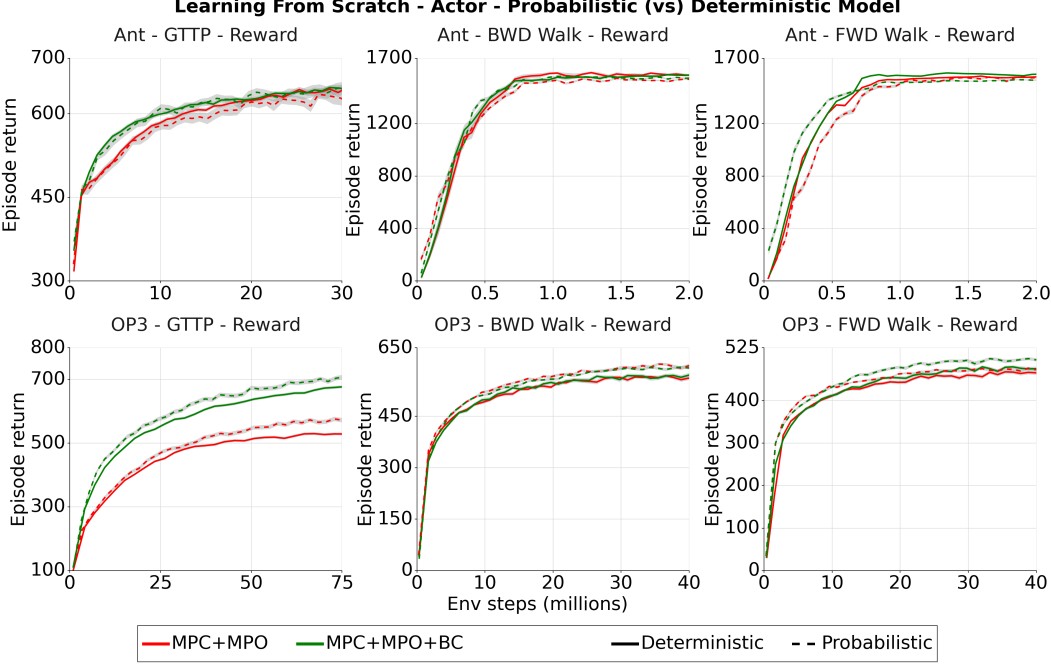

Figure 17: Actor performance of `MPC+MPO` (red) and `MPC+MPO+BC` (green) when trained from scratch using a PETS style stochastic model ensemble (dotted lines) vs the deterministic model (solid lines) used in our main results. There is little difference to using a stochastic ensemble vs a deterministic model on most tasks; on OP3 GTTP using a stochastic model does slightly better but it does not change the results qualitatively (this improvement is reduced when looking at the performance of the amortized proposal; see Fig. 18). Results were obtained using an ensemble of three Gaussian dynamics models for all tasks except the Ant GTTP task where a single Gaussian dynamics model was used. In practice, there is no difference to using a single Gaussian dynamics model vs an ensemble (see Fig. 19 for a comparison of different ensemble sizes on the OP3 GTTP task). Results are averaged over 4 seeds for the GTTP tasks and 2 seeds for the Walking tasks.

Overall, these results validate our choice of deterministic models showing that a single well structured deterministic model can perform as well as stochastic ensemble across challenging locomotion tasks involving high-dimensional, non-linear contact dynamics.

## F.5 RESULTS ON DEEPMIND CONTROL SUITE

In addition to the tasks presented in the paper we also evaluated our approach on externally published and open-sourced locomotion tasks which are a part of the DeepMind Control Suite (Tassa et al., 2018). Specifically, we tested our approach on three locomotion tasks from the DeepMind Control Suite: `stand`, `walk` and `run`, with two embodiments: a 6-DOF `walker` (18-D state space) and a 21-DOF `humanoid` (54-D state space) forming a total of six tasks.

We first tuned MPO (Abdolmaleki et al., 2018b) to match the performance of externally published benchmarks in the literature (Hoffman et al., 2020) across all these tasks. Next, we evaluated our approach of combining MPC with MPO (using a deterministic model) on these tasks; Fig 20 shows the performance of all the algorithmic variants in Sec. 5.2 on these six tasks. All variants perform similarly on the simpler `walker` embodiment; there is no advantage to using MPC across all three tasks for this embodiment. On the `humanoid`, the MPC variants slightly outperform the model-free MPO variants, especially on the challenging `humanoid-run` task. Overall though, the results are broadly similar to those presented in the main text and further lend strength to the central message of the paper; while model-based methods can perform really well on high-dimensional continuous control tasks, well-tuned model-free methods are strong baselines. Only on the most challenging tasks and, especially, in multi-goal/multi-task settings, do model-based methods substantially outperform their model-free counterparts.

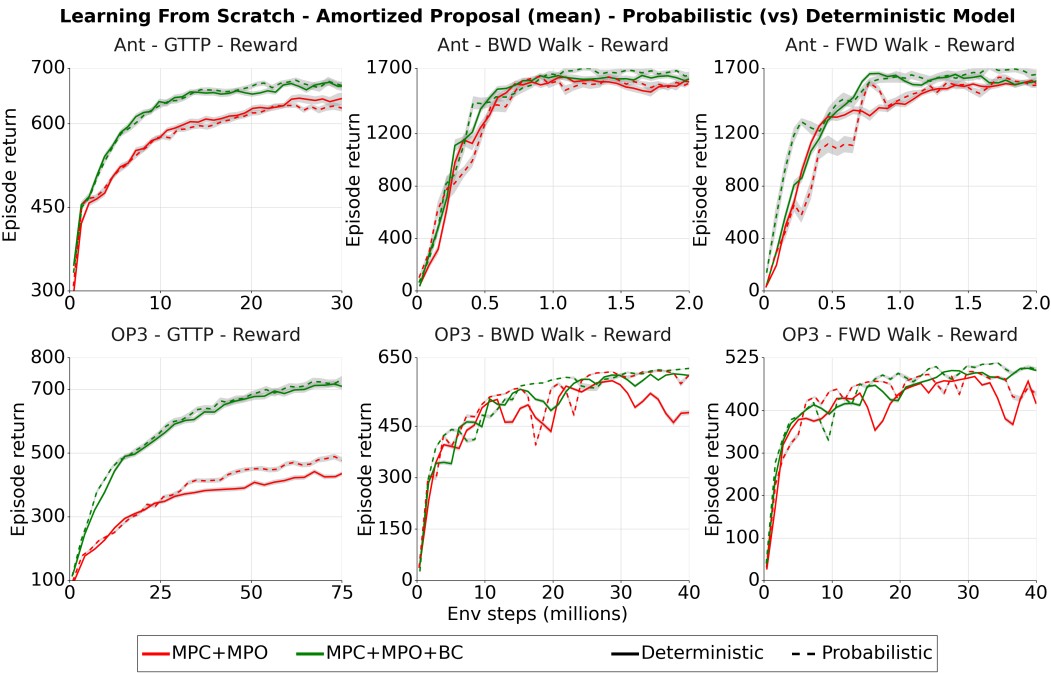

Figure 18: Amortized proposal (mean) performance for `MPC+MPO` (red) and `MPC+MPO+BC` (green), when trained from scratch using a PETS style probabilistic model ensemble (dotted lines) vs the deterministic model (solid lines) used in our main results. Using a stochastic model ensemble instead of a deterministic model does not change the results significantly. Results are averaged over 4 seeds for the GTTP tasks and 2 seeds for the Walking tasks.

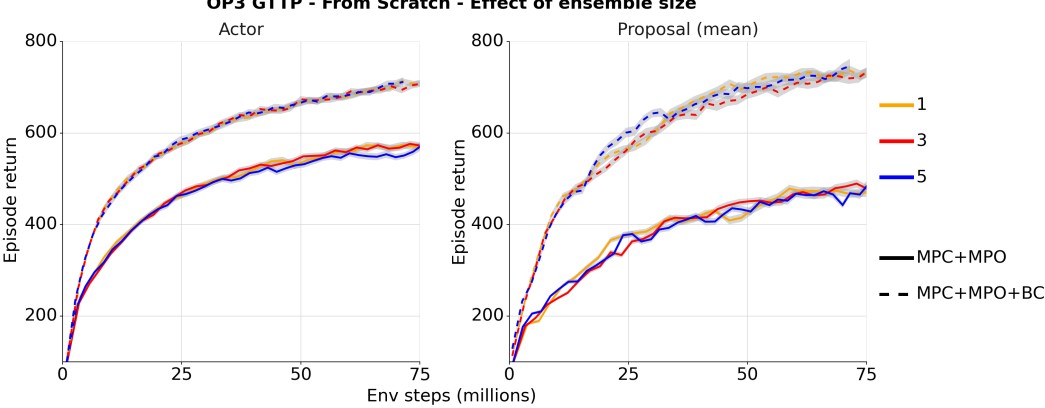

Figure 19: Performance of `MPC+MPO` (solid lines) and `MPC+MPO+BC` (dotted lines) when learning from scratch on the OP3 GTTP task with a PETS style probabilistic model ensemble, where we run a sweep over the size of the ensemble. Using 1, 3 or 5 models in the ensemble does not lead to a significant difference in performance for both the actor (left) and the amortized proposal (right). Results are averaged over four seeds.

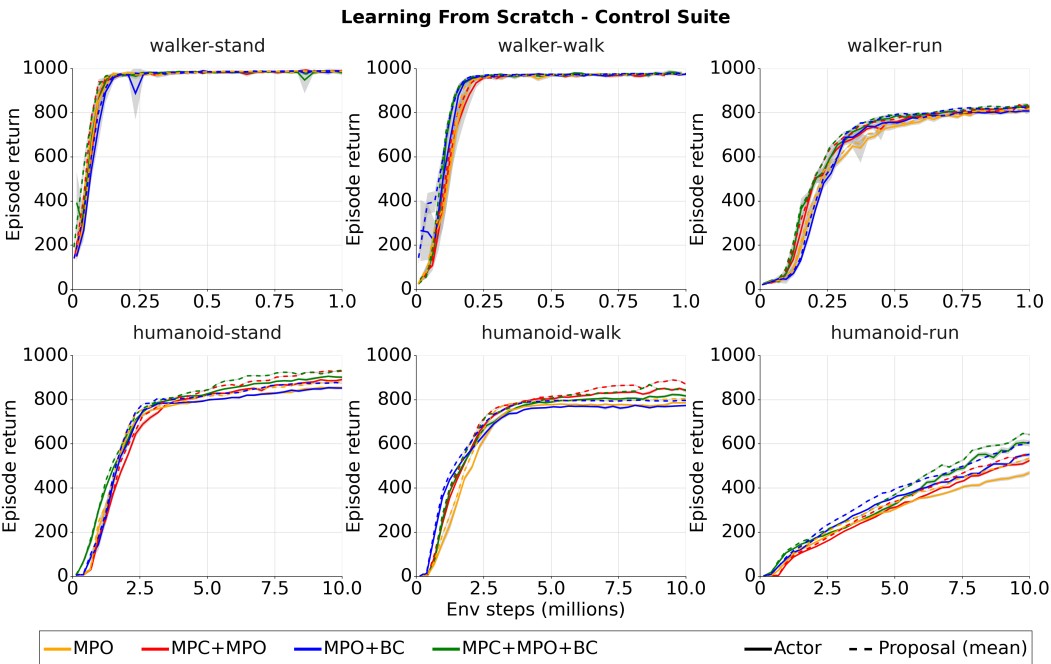

Figure 20: Performance of the algorithmic variants in Sec. 5.2 when learning from scratch on six control suite tasks across two embodiments. On the simpler `walker` embodiment all variants perform similarly. On the challenging `humanoid` embodiment the MPC variants slightly outperform `MPO` especially on the hardest `humanoid-run` task. All results averaged over three seeds, and are the best results from a parameter sweep (see F.2 for parameters). We also swept over bootstrapping from the learned critic while planning for these tasks and found no significant difference with and without bootstrapping; the presented results are without value bootstrapping.

