# OpenReview forum: "Evaluating Model-Based Planning and Planner Amortization for Continuous Control"
_ICLR.cc/2022/Conference — ICLR 2022 Poster_

### Official Review · Reviewer_i15j · 2021-10-29

**Correctness:** 3
**Technical Novelty And Significance:** 2
**Empirical Novelty And Significance:** 2
**Recommendation:** 6
**Confidence:** 4

**Main Review:**


Pros:
1. The quality of the experiment section is very good.
There are proper baselines, and the environments are well designed.
The videos are also very good, providing ideas how well the algorithms are.

The difficulty of the environments are much higher than the ones we currently use for model-based algorithms.
I think they can be good baselines if the code is released along with the algorithm as a default baseline for future research.

2. The related work section is very adequate.

Cons:

1. Some details of the paper are missing.
And since MPO consists of a very important role in the paper, it would be necessary to include a preliminary section.

2. No code is released, and it is almost impossible to reproduce the results presented in the paper.
For the reproductivity, I don't think other researchers can reproduce the algorithms without access to the engineering details.

3. It seems that the most obvious novelty of the paper is the use of MPO as the proposal policy network.
It seems rather arbitrary and I wonder how other model-free algorithms would have fit in the framework.


**Summary Of The Paper:**


In this paper, the authors follow the line of research of adding policy in online model-planning.
They consider multi-task / multi-goal environments, and use MPO as the proposal network, which improves the performance.


**Summary Of The Review:**


I think in general the algorithm is very good. The environments are pretty interesting.
The release of code will be quite important to this project if it aims to encourage future research.

---

> ### Author Response · Authors · 2021-11-21
> **Rebuttal**
>
> We thank the reviewer for their encouraging comments. We are delighted that the reviewer found our experimental section to be of high quality. We share the reviewer’s assessment that some of our environments are substantially harder than the standard environments in the field.
>
> We will address the critical comments below.
>
> > Some details of the paper are missing. And since MPO consists of a very important role in the paper, it would be necessary to include a preliminary section.
>
> It would be helpful if the reviewer can indicate areas where additional detail is necessary. We are very happy to provide additional details where required.
> We have added an additional section to the appendix (Section D) that describes the MPO algorithm in detail.
>
> > It seems that the most obvious novelty of the paper is the use of MPO as the proposal policy network. It seems rather arbitrary and I wonder how other model-free algorithms would have fit in the framework.
>
> We would like to clarify that we do not see the choice of MPO as crucial. In fact, we expect that using any other state-of-the-art off-policy algorithm such as Soft Actor Critic (SAC) should also lead to similar results on the tasks considered.
> Regarding the choice of MPO, we wanted something that worked well in the off policy setting and MPO has been thoroughly evaluated on continuous control tasks and has been shown to have comparable performance to state-of-the-art methods on these tasks [1]. Additionally, MPO’s objective is compatible with the BC loss as they are both log-likelihood based and hence setting any tradeoff between the two can be straightforward. Adding BC loss to MPO has recently also shown interesting results across different tasks [2]. We have also updated the paper with these justifications (see Section D.1 of the appendix).
>
> >No code is released, and it is almost impossible to reproduce the results presented in the paper. For the reproductivity, I don't think other researchers can reproduce the algorithms without access to the engineering details.
>
> We agree that reproducibility is important. To foster more research on the domains presented in this paper we are hoping to release our custom environments. We are not currently planning on releasing the MPO+MPC agents due to infrastructure constraints. However, MPO, the base agent of our work, is open-sourced [1] and MPC as well as BC are well-known algorithmic components [2], so we hope that this work can be reproduced with reasonable effort.
>
> [1] Hoffman, Matt, et al. "Acme: A research framework for distributed reinforcement learning." arXiv preprint arXiv:2006.00979 (2020).
>
> [2] Abdolmaleki et al. On  multi-objective  policy  optimization  as  a  tool  for  reinforcement learning. arXiv  preprint arXiv:2106.08199, 2021.

---

### Official Review · Reviewer_Bjo9 · 2021-11-01

**Correctness:** 4
**Technical Novelty And Significance:** 3
**Empirical Novelty And Significance:** 4
**Recommendation:** 8
**Confidence:** 4

**Main Review:**

Strengths:
- demonstrates the benefit of combined model-learning and proposal learning, in support of MPC, on simple & a challenging GTTP task
- provides a greater understanding about model-transfer and planner-transfer, across tasks
- makes us rethink some of the conventional assumptions about model-based RL

Weaknesses:
- only MPC-settings are considered;  in an ideal world, the reader would also be able
  to understand how these methods stack up against model-based methods that use the model
  in different ways.
- the paper could comment on the possible relative benefits retained by MPC approaches in general,
  over policy-based methods, in terms of adapting to out-of-distribution states.


The paper can in part be seen as an "understanding" paper, one that provides insights into
the interplay between planning and model-free control, and how they can be integrated.
This remains and underexplored area.  Relatedly, the relative benefits of model-learning
in task-specific and for new-tasks with shared dynamics also needs to be better understood,
as noted by the paper.

The paper builds on an interesting parameterized task, the go-to-target-pose (GTTP).
Such parameterized-goal tasks are likely to provide more insight than fixed tasks.

I'd love to see a structured abstract diagram or table that could somehow capture the design space
of model-based planning methods, and the general assumptions that motivate the different design choices.
Caveat:  I don't know if such a diagram is realizable.

re: section 2, other uses of models
Perhaps older refs to backprop-through-time (BPTT) would help communicate that BPTT has been
used even before the rediscovery of backprop, i.e., state adjoints used in control.  Or even just go back
to the Nguyen & Widrow "Truck Backer-upper" from 1990.

"it is possible to train policies on model rollouts to improve data efficiency (Janner et al)"
It might make more sense to simply cite DYNA here, as one among many?

The connections between MPO and SAC could be made more explicit.
Or the differences, if these are relevant for the results presented in the paper.

The idea of separating out proprioceptive information from other goal-related state information
is a good one.  It is in some sense specific to a certain class of MDP, however, so that could
be worthwhile clarifying (although will be obvious to those working on robotics. It also points
to some form of control hierarchy, where new tasks won't need to leverage significantly new proprioceptive states
and actions.  Is the global orientation wrt vertical i.e., basic IMU info, also modeled as being part of the proprioceptive information?
(ok, I now see the answer is "yes" in App A).

Figure 2:  The middle column of graphs, giving Target pose counts, would be better replaced
by simply another task.  The results are highly correlated with the reward graphs in the first column,
and a tired reader will mistake the results as being yet more training curve graphs.

Figure 3:  The overall title of "Model and/or Proposal Transfer -- Actor Performance" is confusing.
And perhaps the legend could be improved, i.e., MO+BC could be relabeled as "MO+BC (no model)"
and "Proposal from scratch" could be relabeled "model transfer"; and "Reloaded Proposal" could become "Proposal Transfer".

It could be Worthwhile noting in the main text that SMC is more conputationally efficient than CEM.
This detail is currently in App E.1, but would be worthwhile being part of the Table 1 discussion.

**Summary Of The Paper:**

The paper provides a thorough, detailed evaluation of model-based RL, as realized via MPC-based planning.
In particular, it examines: (a) how it can be improved via learned policies that act as MPC proposal distributions;
and (b) to understand and disentangle the benefits of (i) model transfer and (ii) proposal transfer,
when using model-based RL. Comparisons are performed using both CEM and SMC samplers.
For the locomotion-related benchmarks considered, MPC with a learned model & proposal can provide modest improvements
wrt a well-tuned model-free baseline (MPO). However, for a more challenging go-to-target-pose (GTTP) task,
they offer significantly improved learning speed and final performance.
The benefits of model-transfer are rather marginal, even when transferred to the same task,
counter-to-common-belief-and-intuition.


**Summary Of The Review:**

This is a reasonably thorough evaluation of common assumptions related to model-based RL, particularly in the context of MPC methods. The ideas are tested on a range of problems, ranging from the simple to the challenging, i.e., the go-to-target-pose (GTTP) task. The documented benefits of learned proposals on difficult tasks such as GTTP, as evaluated using both CEM and SMC, are also a worthwhile contribution.  The empirical work is solid.  A better understanding of model-based methods in continuous control settings is important for the RL community.

---

> ### Author Response · Authors · 2021-11-21
> **Rebuttal**
>
> We thank the reviewer for their encouraging and insightful comments. We share the reviewers view that our paper is aiming to better understand and carefully evaluate common intuitions and assumptions in existing model-based work and that this is important for the model-based RL community. We also share the reviewer’s intuition regarding the parameterized GTTP task and its potential to provide insight for model based and transfer methods in RL. We thank the reviewer for the additional references which we will incorporate in our paper. We address the individual points below.
>
> >only MPC-settings are considered; in an ideal world, the reader would also be able to understand how these methods stack up against model-based methods that use the model in different ways.
>
> While we agree that such a comparison would be great for the field we felt that it would be beyond the scope of a conference paper and a significantly larger project especially given the large number of different approaches. Instead we opted for a careful evaluation in the MPC setting.
> >I'd love to see a structured abstract diagram or table that could somehow capture the design space of model-based planning methods, and the general assumptions that motivate the different design choices. Caveat: I don't know if such a diagram is realizable.
>
> We were unable to clearly realize such a diagram but we have added further experiments to the appendix that may help clarify some of the design choices better. Figs 7-9 present the results of planning with GT dynamics and rewards across two specific axes of choices: 1) the amount of information present in the proposal distribution used for planning, and 2) the computational budget. For the former we considered Gaussian and learned proposals (task-agnostic and fully task-aware) in this work. For the latter, we ran extensive sweeps to quantify planning performance across different settings and chose parameters which best reflected “real-time” control & led to good empirical results. Another meta choice we evaluated in this work was the way in which proposals and models were used within the planner; by considering SMC and CEM planners which both operate in different ways. There are other uses for models beyond model-based planning but this is beyond the scope of this paper.
>
> > re: section 2, other uses of models Perhaps older refs to backprop-through-time (BPTT) would help communicate that BPTT has been used even before the rediscovery of backprop, i.e., state adjoints used in control. Or even just go back to the Nguyen & Widrow "Truck Backer-upper" from 1990.
>
> We thank the reviewer for this reference and have added it to the paper.
>
> > "it is possible to train policies on model rollouts to improve data efficiency (Janner et al)" It might make more sense to simply cite DYNA here, as one among many?
>
> We have added a citation to DYNA in the paper.
>
> >The connections between MPO and SAC could be made more explicit. Or the differences, if these are relevant for the results presented in the paper.
>
> We have added a section to the appendix commenting on our choice of MPO as the learning algorithm (Section D.1). In short, we believe that our results would be similar for the chosen domain if SAC were used as the learning algorithm as opposed to MPO.
> > Figure 2: The middle column of graphs, giving Target pose counts, would be better replaced by simply another task. The results are highly correlated with the reward graphs in the first column, and a tired reader will mistake the results as being yet more training curve graphs.
>
> We added the target pose counts plot to highlight that relatively small differences in (dense) reward in the go to target pose task can correspond to quite different behaviors. We think of the number of target poses as a sparse metric of task success. We have updated the text to clearly highlight that the target pose counts correspond to the reward plots in the first column to reduce confusion for the reader.
> >Figure 3: The overall title of "Model and/or Proposal Transfer -- Actor Performance" is confusing. And perhaps the legend could be improved, i.e., MO+BC could be relabeled as "MO+BC (no model)" and "Proposal from scratch" could be relabeled "model transfer"; and "Reloaded Proposal" could become "Proposal Transfer".
>
> Fig.3 plots the performance of the actor (which uses MPC for all variants except MPO+BC) when a model and/or a proposal are transferred from a source to a target task – hence the title. Renaming “Proposal from scratch” to “Model transfer” would be misleading as there are multiple cases where a model is not transferred (MPO+BC and MPC+MPO+BC) – this is clarified in the main text.

---

> > ### Author Response · Authors · 2021-11-21
> > **Rebuttal part ii**
> >
> > > It could be worthwhile noting in the main text that SMC is more computationally efficient than CEM. This detail is currently in App E.1, but would be worthwhile being part of the Table 1 discussion.
> >
> > While SMC is a non-iterative planner compared to CEM which may need multiple iterations, they use the proposal in different ways. CEM uses the proposal only to generate an initial set of actions while SMC uses the proposal to sample actions at each step of the rollout; this can lead to additional computation overhead for SMC especially if the policy network is large. Additionally, when used in an MPC style loop, a single iteration of CEM might be enough for getting a good action (though this is not the case in our setting). Therefore, in the general case it is hard to argue that SMC is computationally efficient than CEM; as a consequence we primarily argue that our choice of SMC is motivated by the fact that it uses the proposal better than CEM.

---

> > > ### Comment · Reviewer_Bjo9 · 2021-11-30
> > > **acknowledgement of rebuttal comments**
> > >
> > > Thank you to the authors for the rebuttal comments, which address my concerns.  I remain very positive about this paper, and think that it makes a solid contribution to ICLR, in an important direction.

---

### Official Review · Reviewer_54KF · 2021-11-02

**Correctness:** 4
**Technical Novelty And Significance:** 1
**Empirical Novelty And Significance:** 2
**Recommendation:** 6
**Confidence:** 3

**Main Review:**

This paper is very clearly written, well argued, and well executed.    The experiments are well-designed and clearly illuminate the points being made.  The paper adds an increment to the collective knowledge of the field about RL methods.

I also find it deeply unexciting and not at all imaginative.    I would have been more enthusiastic about a less well executed paper with a really new idea.   Surely the fact that transfer isn't working very well tells us that we need some new insights (or I guess, an argument about why all of our intuition about transfer in problems like these is wrong).   Is there a way to improve the planner so that it's less stupid, for example by learning landmarks or making it hierarchical?  Could we make the proposal distribution depend on the goal in the GTTP tasks?

I have some small comments/questions, but none of them are critical:
- In the early parts of the paper it would be good to clarify what you mean by "learning proposals" (it's made clear later)
- I found it jarring that the word "proposal" is used frequently when really "proposal distribution" is what is intended;  to me a "proposal" would be a single sample from a "proposal distribution."
- When you discuss "target poses" do you really mean "target configurations"?   (This is ambiguous, I guess, but in robotics it's most common to use "pose" for a 6DOF pose of a rigid object and "configuration" for anything more complicated.
- Do you think anything different would have happened if the goals were a bit more naturalistic (e.g., get the centroid of the agent into a region)---it's rare in any natural situation or application that the entire configuration of a complex agent would need to be copied.
- The design of the model was interesting---I was eager to read more about it in the paper body.
- I am concerned that stochasticity was "waved away" without enough consideration.  In many important domains, the distribution of possible outcomes is not at all well summarized by the mean.  It seems important to at least acknowledge this.
- It was a bit confusing to read the phrase "task-agnostic proposal pre-trained with a behavioral-cloning objective."   It's hard to imagine a situation in which behavioral-cloning is task-agnostic unless, for example, you train on BC data from a whole mixture of tasks.  Perhaps that's what you did.

(To the degree that I lack confidence it is with respect to the detailed novelty of this paper---I don't keep up with this sub-part of the literature and so I can't attest to whether or not this was already all well-known, for example.  But I think I understand the methods described at a sufficient level and get the experiments and their overall message.)

**Summary Of The Paper:**

This paper performs a careful study of the advantages of integrating model-based components into an RL system.  In particular, they clearly distinguish between the effects of learning a predictive model and learning a proposal distribution for a forward-sampling planner.   They sensibly focus on what seems like the "sweet spot" for model-based methods:  transfer of a model from one task to another in a domain with the same basic dynamics.   They find that, when using forward-sampling planners, the proposal distribution is the driving factor of success, so that learning and transferring a good model does not, in itself, improve performance much.

**Summary Of The Review:**

Whether or not to accept this paper depends on what we think conference publications are for.

This paper contributes some clear and well justified knowledge.  But the increment is small and it seems likely to me that substantially new and different techniques will arise soon and render this irrelevant.    It seems unlikely that this paper will inspire or excite new research.

Ultimately, this decision is above my pay grade, and depends on the program chairs' views about the role of paper acceptance.

My guess is that a good but not hugely exciting paper is thought to be on the positive side, so I'll lean positive.

---

> ### Author Response · Authors · 2021-11-21
> **Rebuttal**
>
> Different researchers set out with fundamentally different goals.  While some people find it most exciting to work on novel ideas (that may potentially be evaluated in specific settings only), we do not believe that this approach alone is enough to advance a maturing field.  For researchers across time and institutions to truly build on the work of one another, thorough investigations of clean approaches are necessary.  Such thoroughness is how we understand the limits of approaches and how we see clearly where innovations are most likely to help.
>
> Novel ideas are obviously important but while there has been a significant focus on novelty in recent ML / RL literature, more needs to be done on careful analysis of the strengths and weaknesses of existing approaches. Progress depends not only on novelty but also on carefully understanding whether an idea is useful, where it fails, and why. This is the primary focus of this paper; specifically we want to build intuition on model-based approaches [e.g. 1-5] and thoroughly validate claims relating to their data efficiency and transfer capabilities on challenging continuous control environments under specific assumptions such as access to the true reward function. Contrary the results from prior work [e.g 1-5] our insights suggest that there are clear regimes and task settings where model based planning (without a proposal) struggles significantly, model-free approaches can be as data efficient as model-based methods, and transferring a model is not by itself enough to improve data efficiency in challenging tasks. These results are surprising and contrary to widely held beliefs in the community. We believe our insights will be useful for the field in general; our results also highlight specific domains and paradigms where model-based methods can significantly outperform model-free methods (e.g. on multi-task/multi-goal domains) which provides important guidance for future work in model-based RL.
>
> > It seems unlikely that this paper will inspire or excite new research.
>
> Without a clear consensus of what the current methods can do, when pushed to their limits, most people find it difficult to decide where to attempt to innovate.  Simply attempting to establish a clear state of the world for the current methods is actually a tremendously ambitious goal that requires thoroughness.
>
> > Is there a way to improve the planner so that it's less stupid, for example by learning landmarks or making it hierarchical?
>
> As stated above, the primary focus of the paper is about building insights on model-based planning methods and validating commonly held beliefs about such methods. As our results show, model-based planning methods can struggle to significantly outperform model-free methods on certain task regimes and domains. To tackle this, we do the obvious things to get planners to work (e.g. via the use of learned goal conditioning proposals), and show that this can help on the hardest of tasks considered. We believe it is more important to show that there is a need for improved planners (there are clear limitations to existing approaches) as a first step,  instead of directly jumping to proposing a more complicated solution for something for which the need is unclear.
>
> As the reviewer suggests, a particular option to improve planning methods is to consider hierarchical variants. While there is already a growing literature on hierarchical planning, the right way to do it for embodied motor control tasks has not been established, and indeed it is not even clear as of yet which tasks will show benefit from hierarchical planning relative to flat planning.  We find that problem space compelling, but it is somewhat flippant and unreasonable to dismiss our paper on the grounds that it isn't a totally different paper solving a substantially harder problem that remains open in the field.
> >Could we make the proposal distribution depend on the goal in the GTTP tasks?
>
> We do condition proposals on the goals in certain experiments in our paper. In our experiments from scratch (Section 5.2 of the main paper), the learned proposal distribution does depend on the goal. Only the proposal distribution that is transferred from a source task to target task in our “proposal transfer” experiments (Section 5.4) is “goal unaware” –  there is no general way to transfer a goal aware proposal in our setting (as a goal-aware proposal would effectively solve the problem completely negating a need for transfer).

---

> > ### Author Response · Authors · 2021-11-21
> > **Rebuttal part ii**
> >
> >
> > >In the early parts of the paper it would be good to clarify what you mean by "learning proposals" (it's made clear later)
> >
> > We have clarified the use of the term proposal in the early part of the paper.
> >
> > >I found it jarring that the word "proposal" is used frequently when really "proposal distribution" is what is intended; to me a "proposal" would be a single sample from a "proposal distribution."
> >
> > We have added a footnote where the term “proposal” first appears in the introduction to convey that we are referring to “proposal distributions”, which are distributions over actions, when we say “proposal”.
> > > When you discuss "target poses" do you really mean "target configurations"? (This is ambiguous, I guess, but in robotics it's most common to use "pose" for a 6DOF pose of a rigid object and "configuration" for anything more complicated.
> >
> > Yes, we do mean “target configurations” (of all the joints of the robot) when referring to “target poses”. When referring to people (and humanoid like bodies), pose is frequently used to refer to the body configuration. E.g. pose estimation is commonly used to refer to estimation of a human's entire body. We follow this convention in this paper.
> >
> > >Do you think anything different would have happened if the goals were a bit more naturalistic (e.g., get the centroid of the agent into a region)---it's rare in any natural situation or application that the entire configuration of a complex agent would need to be copied.
> >
> > We believe that similar trends would hold for other multi-task/multi-goal settings. The more broad the goal distribution and the harder the task complexity the more we would expect model-based planning with a learned model to improve upon model-free agents in terms of data efficiency.
> >
> > >I am concerned that stochasticity was "waved away" without enough consideration. In many important domains, the distribution of possible outcomes is not at all well summarized by the mean. It seems important to at least acknowledge this.
> >
> > The experiments in our paper are in deterministic environments. We primarily mentioned stochastic models (and ensembles) as they are used in some of the related literature (see [1-5] below). We agree with the reviewer that in truly stochastic environments stochastic models should be used.
> > To ensure that our results are robust to the choice of model determinism we did run additional experiments with stochastic models and ensembles of stochastic models (similar to prior work from Chua et al. [2]). These results are presented in Section F.4 and Figures 17-19 of the appendix. In practice, on the environments considered we did not see a significant difference between deterministic vs stochastic models. On the contrary, if the target MDPs had stochasticity then deterministic models would not be a good choice – but as shown, our approach can be easily extended to use stochastic models to handle this setting as well.
> >
> > >It was a bit confusing to read the phrase "task-agnostic proposal pre-trained with a behavioral-cloning objective." It's hard to imagine a situation in which behavioral-cloning is task-agnostic unless, for example, you train on BC data from a whole mixture of tasks. Perhaps that's what you did.
> >
> > In our setting the task-agnostic proposal does not have access to the goal observations. For example, in the GTTP task the task-agnostic proposal does not have access to the target pose. In this setting, there are several different target poses the agent has to reach and the task agnostic proposal is now trained via BC on data with all these poses as targets – this situation is exactly what the reviewer has pointed out, we train the proposal via BC on data from a whole mixture of tasks.
> >
> > [1] N. Heess, G. Wayne, D. Silver, T. Lillicrap, Y. Tassa, and T. Erez. Learning continuous control policies by stochastic value gradients. Neural Information Processing Systems (Neurips), 2015.
> >
> > [2] K. Chua, R. Calandra, R. McAllister, and S. Levine. Deep reinforcement learning in a handful of trials using probabilistic dynamics models. Neural Information Processing, 2018
> >
> > [3] M. Janner, J. Fu, M. Zhang, and S. Levine. When to trust your model: Model-based policy optimization. Neural Information Processing Systems, 2019.
> >
> > [4] A. Nagabandi, K. Konolige, S. Levine, and V. Kumar. Deep dynamics models for learning dexterous manipulation. In Conference on Robot Learning, 2020.
> >
> > [5] J. Buckman, D. Hafner, G. Tucker, E. Brevdo, and H. Lee. Sample-efficient reinforcement learning with stochastic ensemble value expansion. arXiv preprint arXiv:1807.01675, 2018.
> >
> > [6] Lutter, Michael, Leonard Hasenclever, Arunkumar Byravan, Gabriel Dulac-Arnold, Piotr Trochim, Nicolas Heess, Josh Merel, and Yuval Tassa. 2021. “Learning Dynamics Models for Model Predictive Agents.” arXiv [cs.LG]. arXiv. http://arxiv.org/abs/2109.14311.

---

> > > ### Comment · Reviewer_54KF · 2021-11-29
> > > **Thanks for your reply.**
> > >
> > > Thanks for your reply!

---

### Official Review · Reviewer_CeZd · 2021-11-03

**Correctness:** 4
**Technical Novelty And Significance:** 2
**Empirical Novelty And Significance:** 3
**Recommendation:** 6
**Confidence:** 4

**Main Review:**

Strengths
--
- The paper investigates a topic that is interesting and timely
- The results illustrate settings in which the quality of the policy proposals are very important to the final performance
- The proposed hybrid method is straightforward
- The paper includes exhaustive ablations

Weaknesses
--
- W1 Some writing issues: **W1.1 Missing proposal terminology** and **W1.2 Dubious justification of model determinism**
- W2 The presented evidence doesn't fully validate the claims: **W2.1 Missing evidence of computational burden**,  **W2.2 Narrow set of environments used**, and **W2.3 Narrow set of tasks used**

Weakness 1:
--
-  **W1.1 Missing proposal terminology** The terminology "proposal" used in the abstract and introduction needs a clear definition if it's going to be used here. Is it a single action sampled from a policy? Is it a stochastic policy? Unfortunately, the current introduction forces the reader to guess.
- **W1.2 Dubious justification of model determinism** S4.2 The use of deterministic models here is fine, but I'd argue that it's a reasonable assumption mainly because of determinism (I assume) in the true environment dynamics. If the MDPs investigated had significant stochasticity in the environment dynamics, then perhaps it would not be a great assumption, and performance would be significantly worse. S4.2 should be explicit about the degree of stochasticity in the actual environment dynamics. If the degree is insignificant (or none), it'd be great to get the authors' take on whether they agree that the main reason deterministic dynamics models work well is simply that the modeling class can represent the true dynamics models employed in the target MDPs.

Weakness 2:
--
- **W2.1 Missing evidence of computational burden** S4.3 The claim that planning in these tasks cannot be performed alone with the ground-truth dynamics model and reward function requires more detail and requires more evidence. It is fair to assume that with enough search time/compute, a long-enough planning horizon, and access to ground-truth dynamics and rewards, the tasks can be solved e.g. via brute force search. Thus, the claim needs to be more specific about the computational restrictions that necessitate the use of a proposal-generating policy. This issue also appears in the first paragraph of S6 -- there's no actual discussion or experimentation of what constitutes 'limited' computation time. Because this claim is central to the paper's message -- that a good dynamics model is insufficient for transfer -- the paper would be much stronger if it presented evidence across a clear regime of computational time constraints that the result holds. For example, a plot of Performance vs. Computation time, with the 'optimal' and 'best proposal-based method' performance as single points (or sets of points), accompanied by the performance of MPC with GT dynamics and rewards with different computational budgets plotted as a curve. By the paper's claims, the latter MPC curve should be significantly below the 'optimal' and 'best proposal-based method' for reasonable computational regimes, and above the 'best proposal-based method' when an 'unreasonable' amount of compute is assumed (if feasible).
- **W2.2 Narrow set of environments used** The employed set of multi-task evaluation environments in the paper is quite small. While it's good that the single-task environments are presented in the appendix, without a broader set of multi-task environments, we cannot draw broad conclusions about these methods in other multi-task settings
- **W2.3 Narrow set of tasks used** The multi-task evaluation is limited to tasks in which the multi-task nature is target-goal reaching. Results presented on different MDPs in which the reward function variation is more sophisticated than goal reaching would provide more support for the claims.

Minor weaknesses
--
- The DAGGER reference in 4.1 is too tangential (neither necessary nor particularly useful).
- S4.3 Near "We can then improve our proposal by minimizing the KL divergence", the optimization variables need to be clarified, since it's unclear if the $\pi_\theta$ contained within $\tilde{\pi}_{\mathcal B}$ is optimized. However, Eq. 1 clarifies this, so perhaps just call it a 'forward' KL divergence.
- S4.1 says more details about the planner are in Sec B of the supplement, but the supplement didn't contain a .pdf (I found Sec B in the appendix of the .pdf)
- Fig 2 needs more details in the caption or in the main text to describe the difference between 'actor' and 'proposal'. My understanding is that the 'actor' is the MPC actor if MPC is used, or equivalent to the policy (and the 'proposal') if MPC is not used.
- Footnote 2 requires a citation or evidence for the claim "MPO has a known problem with shrinking policy variances". Furthermore, this hypothesis could be tested simply by adding a policy-entropy bonus to objective function, no?


**Summary Of The Paper:**

This paper's goal is to evaluate the sample-efficiency of planning-based and policy-based approaches, as well as hybrid versions of each, in several multi-task continuous control environments. The paper shows that for the tasks investigated and the specific computational budget allocated to the planner, the planner performs significantly worse without a good proposal distribution (policy sampler), even when employed with ground-truth access to dynamics and rewards. Then, the paper presents multi-task experiments with variations of their proposed hybrid planning-amortized method that concurrently learns a proposal distribution, model, and critic; these experiments show that the MPC-based variants outperformed a model-free method to varying degrees.

**Summary Of The Review:**

While the research question and initial evidence are promising, the main issue with the paper is Weakness 2 above, particularly W2.1. Evidence is required to support the claimed computational burden of planning with ground-truth dynamics model and rewards. Furthermore, the multi-task experimental results are only conducted in two environments (W2.2) and only for goal-directed tasks (W2.3), which doesn't support the claim that the quality of the proposal distribution is critical to planning in *general* **multi-task** settings; it instead only constitutes evidence that the quality of the proposal distribution is critical to planning in *some* **goal-directed tasks**.

In my opinion, this missing evidence weakens the paper slightly below the bar of acceptability.

---

> ### Author Response · Authors · 2021-11-21
> **Rebuttal**
>
> We thank the reviewer for their comments and suggestions. We address the critical comments below.
>
> > W1.1 Missing proposal terminology
>
> We have changed the abstract to read “proposal distributions” instead of “proposals”, and we have added a footnote where the term “proposal” first appears in the introduction to convey that we are referring to “proposal distributions”, which are distributions over actions when we say “proposal”.
> > W1.2 Dubious justification of model determinism
>
> We agree with the reviewer that the environments we consider are deterministic and thus a deterministic model may suffice. However, prior work uses stochastic models (and ensembles) to better capture modeling uncertainty in similarly deterministic environments (see [1-5] below) . This is why we felt it was important to justify our choice of deterministic models.
> To ensure that our results are robust to the choice of model determinism we did run additional experiments with stochastic models and ensembles of stochastic models (similar to prior work from Chua et al. [2]). These results are presented in Section F.4 and Figures 17-19 of the appendix. In practice, on the environments considered we did not see a significant difference between deterministic vs stochastic models. We do agree with the reviewer that if the target MDPs had stochasticity then deterministic models would not be a good choice – but as shown, our approach can be easily extended to use stochastic models to handle this setting as well.
> >W2.2 Narrow set of environments used and W2.3 Narrow set of tasks used (need broader set of multi task environments and rewards other than goal-reaching)
>
> While we agree that an even larger number of environments and tasks would be desirable in principle we feel that we have performed a careful evaluation in a wide range of environments and tasks. Adding even more environments would be beyond the scope of a conference paper and quite time consuming and expensive. In our view it would only marginally improve the paper.
>
> >W2.1 Missing evidence of computational burden.
>
> We thank the reviewer for raising these points. We agree that, in principle, with enough search time, a long enough horizon, ground-truth dynamics and known rewards these tasks can be solved. However, the compute required depends on the complexity of the task and the dimensionality of the action space. We did run experiments with a wide range of computational budgets and planning horizons and have added plots of the performance vs computation time (as requested by the reviewer) when running MPC with GT rewards and dynamics and Gaussian/learned task-agnostic proposal for most of the locomotion tasks in the paper in section F.1 of the appendix (Figs 7-9). We also additionally show performance of the proposals themselves (without planning) and the results from the best performing RL agent for comparison.
>
> To summarize the results, on the ant tasks, using more samples and longer horizons improves performance for both a standard Gaussian proposal and a pre-trained task-agnostic proposal, though the former performs much worse than the latter on the walking tasks and performs similarly on the GTTP task. The results for the OP3 GTTP task are qualitatively different. Whereas performance for the pre-trained proposal improves with more samples, we did not see any task progress for the standard Gaussian proposal (even with 25000 samples, 2 orders of magnitude compared to what we use in the learning experiments). This points to the difficulty of sample-based search in high-dimensional settings, especially in the presence of termination conditions. [6] conduct experiments with ground-truth planning on control suite tasks and also find that ground truth planning can fail completely in high-dimensional humanoid tasks. Overall, just planning with either proposal does significantly worse than our best performing RL agent, highlighting the need for learning good proposals for MPC.
>
> With regards to our choice of computational budget for the learning experiments we were interested in a budget that could be eventually used for real-time control (at 30-50Hz). These computational considerations led us to use 250 samples and a planning horizon of 10. We did ablate this choice in our set of experiments (see Figs 10 and 11 of the appendix); increasing either the horizon or the number of samples did not make a difference with respect to our results.

---

> > ### Author Response · Authors · 2021-11-21
> > **Rebuttal part ii**
> >
> > We have also addressed the various minor comments.
> >
> > >S4.3 Near "We can then improve our proposal by minimizing the KL divergence", the optimization variables need to be clarified. However, Eq. 1 clarifies this, so perhaps just call it a 'forward' KL divergence.
> >
> > We have changed this to “minimizing the forward KL divergence”.
> > >S4.1 says more details about the planner are in Sec B of the supplement, but the supplement didn't contain a .pdf (I found Sec B in the appendix of the .pdf)
> >
> > We have changed this in the main text to “See Sec.B for more details”, and have linked Sec.B to the corresponding section of the appendix.
> > >Fig 2 needs more details in the caption or in the main text to describe the difference between 'actor' and 'proposal'. My understanding is that the 'actor' is the MPC actor if MPC is used, or equivalent to the policy (and the 'proposal') if MPC is not used.
> >
> > We have updated the captions of Fig. 2 to explain the difference between the “Actor” and “Proposal”.
> > >Footnote 2 requires a citation or evidence for the claim "MPO has a known problem with shrinking policy variances". Furthermore, this hypothesis could be tested simply by adding a policy-entropy bonus to objective function, no?
> >
> > We thank the reviewer for pointing this out. We have modified the footnote and added a citation. It now reads: “Recent work has shown that similar combinations of MPO and a BC objective work quite well across challenging online and offline-RL settings [7].”
> >
> > [1] N. Heess, G. Wayne, D. Silver, T. Lillicrap, Y. Tassa, and T. Erez. Learning continuous control policies by stochastic value gradients. Neural Information Processing Systems (Neurips), 2015.
> >
> > [2] K. Chua, R. Calandra, R. McAllister, and S. Levine. Deep reinforcement learning in a handful of trials using probabilistic dynamics models. Neural Information Processing, 2018.
> >
> > [3] M. Janner, J. Fu, M. Zhang, and S. Levine. When to trust your model: Model-based policy optimization. Neural Information Processing Systems, 2019.
> >
> > [4] A. Nagabandi, K. Konolige, S. Levine, and V. Kumar. Deep dynamics models for learning dexterous manipulation. In Conference on Robot Learning, 2020.
> >
> > [5] J. Buckman, D. Hafner, G. Tucker, E. Brevdo, and H. Lee. Sample-efficient reinforcement learning with stochastic ensemble value expansion. arXiv preprint arXiv:1807.01675, 2018.
> >
> > [6] Lutter, Michael, Leonard Hasenclever, Arunkumar Byravan, Gabriel Dulac-Arnold, Piotr Trochim, Nicolas Heess, Josh Merel, and Yuval Tassa. 2021. “Learning Dynamics Models for Model Predictive Agents.” arXiv [cs.LG]. arXiv. http://arxiv.org/abs/2109.14311.
> >
> > [7] Abdolmaleki et al. On  multi-objective  policy  optimization  as  a  tool  for  reinforcement learning. arXiv  preprint arXiv:2106.08199, 2021.

---

> > > ### Comment · Reviewer_CeZd · 2021-11-30
> > > **Reviewer response to rebuttal**
> > >
> > > I thank the authors for their response -- it has addressed my concerns. I will increase my recommendation score.

---

### Author Response · Authors · 2021-11-29
**Rebuttal follow-up**

We would appreciate hearing back from the reviewers on whether their concerns have been addressed by our responses and updated paper. We are happy to answer any remaining questions. Thank you.

---

### Decision · Program_Chairs · 2022-01-20

**Decision:**

Accept (Poster)

**Comment:**

The paper examines the advantage of using models in RL.  The authors' rebuttals convinced us of the value of the paper.